# Deep Ensembling with No Overhead for either Training or Testing: The All-Round Blessings of Dynamic Sparsity

**Shiwei Liu[1], Tianlong Chen[2], Zahra Atashgahi[3], Xiaohan Chen[2], Ghada Sokar[1],**
**Elena Mocanu[3], Mykola Pechenizkiy[1], Zhangyang Wang[2], Decebal Constantin Mocanu[1,3]**
[1]Eindhoven University of Technology, [2]University of Texas at Austin, [3]University of Twente,
`{s.liu3,g.a.z.n.sokar,m.pechenizkiy}@tue.nl`
`{tianlong.chen,xiaohan.chen,atlaswang}@utexas.edu`
`{z.atashgahi,e.mocanu,d.c.mocanu}@utwente.nl`

## Abstract

The success of deep ensembles on improving predictive performance, uncertainty estimation, and out-of-distribution robustness has been extensively studied in the machine learning literature. Albeit the promising results, naively training multiple deep neural networks and combining their predictions at inference leads to prohibitive computational costs and memory requirements. Recently proposed efficient ensemble approaches reach the performance of the traditional deep ensembles with significantly lower costs. However, the training resources required by these approaches are still at least the same as training a single dense model. In this work, we draw a unique connection between sparse neural network training and deep ensembles, yielding a novel efficient ensemble learning framework called $FreeTickets$. Instead of training multiple dense networks and averaging them, we directly train sparse subnetworks from scratch and extract diverse yet accurate subnetworks during this efficient, sparse-to-sparse training. Our framework, $FreeTickets$, is defined as the ensemble of these relatively cheap sparse subnetworks. Despite being an ensemble method, $FreeTickets$ has even fewer parameters and training FLOPs than a single dense model. This seemingly counter-intuitive outcome is due to the ultra training/inference efficiency of dynamic sparse training. $FreeTickets$ surpasses the dense baseline in all the following criteria: *prediction accuracy, uncertainty estimation, out-of-distribution (OoD) robustness, as well as efficiency for both training and inference.* Impressively, $FreeTickets$ outperforms the naive deep ensemble with ResNet50 on ImageNet using around only 1/5 of the training FLOPs required by the latter. We have released our source code at `https://github.com/VITA-Group/FreeTickets`.

## 1 Introduction

Ensembles (Hansen & Salamon, 1990; Levin et al., 1990) of neural networks have received large success in terms of the predictive accuracy (Perrone & Cooper, 1992; Breiman, 1996; Dietterich, 2000; Xie et al., 2013), uncertainty estimation (Fort et al., 2019; Lakshminarayanan et al., 2017; Wen et al., 2020; Havasi et al., 2021), and out-of-distribution robustness (Ovadia et al., 2019a; Gustafsson et al., 2020). Given the fact that there are a wide variety of local minima solutions located in the high dimensional optimization landscape of deep neural networks and various randomness (e.g., random initialization, random mini-batch shuffling) occurring during training, neural networks trained with different random seeds usually converge to different low-loss basins with similar error rates (Fort et al., 2019; Ge et al., 2015; Kawaguchi, 2016; Wen et al., 2019). Deep ensembles, combining the predictions of these low-loss networks, achieve large performance improvements over a single network (Huang et al., 2017; Garipov et al., 2018; Evci et al., 2020b).

Despite the promising performance improvement, the traditional deep ensemble naively trains multiple deep neural networks independently and ensembles them, whose training and inference cost increases linearly with the number of the ensemble members. Recent works on efficient ensembles are able

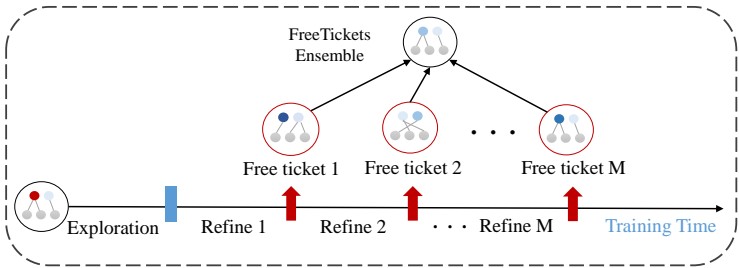

Figure 1: Illustration of $FreeTickets$ with EDST Ensemble as an example. EDST Ensemble, consisting of one exploration phase and M sequential refinement phases, produces M diverse subnetworks with very low cost (hence called "free tickets"). By combining all these free tickets, EDST Ensemble matches the performance of the dense ensemble with only half of FLOPs required to train a single dense model.

to reach the performance of dense ensembles with negligible overhead compared to a single dense model (Wen et al., 2020; Wenzel et al., 2020; Havasi et al., 2021). However, the training resources required by these approaches are still at least the same as training a single dense model. Since the size of advanced deep neural networks is inevitably exploding (Touvron et al., 2020; Dosovitskiy et al., 2021; Brown et al., 2020; Touvron et al., 2021), the associated enormous training costs are potentially beyond the reach of most researchers and startups, leading to financial and environmental concerns (García-Martín et al., 2019; Schwartz et al., 2019; Strubell et al., 2019).

On the other hand, researchers have recently explored the possibility of directly training sparse neural networks from scratch (Mocanu et al., 2016; Liu et al., 2020a; Evci et al., 2019), while trying to maintain comparable performance. Training a sparse network from scratch typically results in worse performance than the traditional network pruning (Kalchbrenner et al., 2018; Evci et al., 2019), with the exception of Dynamic Sparse Training (DST) (Mocanu et al., 2018; Evci et al., 2020a; Liu et al., 2021c;a). Instead of inheriting weights from dense networks, DST starts from a randomly-initialized sparse network and optimizes the model weights together with the sparse connectivity during training. However, the current only way for DST to match the performance of its dense counterpart on the popular benchmark, e.g., ResNet-50 on ImageNet, is to extend the training time (Evci et al., 2020a), which might require thousands of training epochs for extremely sparse models (Liu et al., 2021c).

In this paper, we attempt to address the above-mentioned two challenges jointly by drawing a unique connection between sparse training and deep ensembles. Specifically, we ask the following question:

*Instead of allocating all resources to find a strong winning ticket, can we find many weak tickets with very low costs (free tickets), such that the combination of these free tickets can significantly outperform the single dense network, even the dense ensemble?*

Note that it is not trivial to obtain free tickets. To guarantee superior ensemble performance, three key desiderata that the free tickets are expected to satisfy (1) *high diversity*: according to the ensemble theory (LeCun et al., 2015; Hansen & Salamon, 1990; Ovadia et al., 2019b), higher diversity among ensemble members leads to higher predictive performance; (2) *high accessibility*: free tickets should be cheap to obtain so that the overall training cost does not compromise too much; and (3) *high expressibility*: the performance of each free ticket should be comparable with the dense model.

Leveraging the insight from Liu et al. (2020b) that a full network contains a plenitude of performative subnetworks that are very different in the topological space, we introduce the concept of $FreeTickets$, an efficient ensemble framework that utilizes sparse training techniques to create cheap yet accurate subnetworks for ensemble. Furthermore, we instantiate $FreeTickets$ by proposing two efficient ensemble methods – Dynamic Sparse Training Ensemble (DST Ensemble) and Efficient Dynamic Sparse Training Ensemble (EDST Ensemble). Both methods yield diverse subnetworks that consummately satisfy the above-mentioned criteria. We summarize our contributions below:

- Our first method, DST Ensemble, independently trains multiple subnetworks from scratch with dynamic sparsity. By averaging the predictions of these subnetworks, DST Ensemble improves the predictive accuracy, OoD robustness, uncertainty estimation, and efficiency over the traditional dense ensemble.
- Our second, light-weight method ( EDST Ensemble) yields many free tickets in **one single run**, which is more efficient to train and test than a **single dense model**, while approaching the performance of the traditional **dense ensemble**.

- We analyze the diversity of the individual subnetworks generated by our methods and confirm the effectiveness of our methods on inducing model diversity.
- Our results suggest that besides the training/inference efficiency, sparse neural networks also enjoy other favorable properties which are absent in dense networks (robustness, out-of-distribution generalization, etc), opening the path for new research directions.

## 2 RELATED WORKS

**Efficient Ensembles.** One major limitation of ensembles is the expensive computational and memory costs for both training and testing. To address this problem, various approaches have been proposed. TreeNet (Lee et al., 2015) shares weights in earlier layers and splits the following model into several branches, improving accuracy over the dense ensemble. Monte Carlo Dropout (Gal & Ghahramani, 2016) can be used to approximate model uncertainty in deep learning without sacrificing either computational complexity or test accuracy. BatchEnsemble (Wen et al., 2020) was proposed to improve parameter efficiency by decomposing the ensemble members into the product of a shared matrix and a rank-one matrix personalized for each member. MIMO (Havasi et al., 2021) uses a multi-input multi-output configuration to concurrently discover subnetworks that co-habit the dense network without explicit separation. Snapshot (Huang et al., 2017) and FGE (Garipov et al., 2018) discover diverse models by using cyclical learning rate schedules. Furlanello et al. (2018) applied knowledge distillation (Hinton et al., 2015) to train several generations of dense students. The ensemble of the dense students outperforms the teacher model significantly. Other related works include but are not limited to hyper-batch ensembles (Wenzel et al., 2020) and Late Phase (Oswald et al., 2021). However, the training resources required by these methods are still at least the same as training a single dense model. In contrast to the existing efficient ensemble methods, our methods (EDST Ensemble) can match the performance of naive ensemble with only a fraction of the resources required by training a single dense network.

**Dynamic Sparse Training.** Dynamic Sparse Training (DST) is a class of methods that enables training sparse neural networks from scratch by optimizing the sparse connectivity and the weight values simultaneously during training. DST stems from Sparse Evolutionary Training (SET) (Mocanu et al., 2018), a sparse training algorithm that outperforms training a static sparse model from scratch (Mocanu et al., 2016; Evci et al., 2019). Weight reallocation was further proposed in Mostafa & Wang (2019); Dettmers & Zettlemoyer (2019); Liu et al. (2021b) to reallocate new weights across layers for better layer-wise sparsity. Further, Dettmers & Zettlemoyer (2019); Evci et al. (2020a) leverage the gradient information in the backward pass to guide the optimization of sparse connectivity and demonstrate substantial performance improvement. Some recent works (Jayakumar et al., 2020; Raihan & Aamodt, 2020; Liu et al., 2021c) demonstrate that a large range of exploration in the parameter space is important for dynamic sparse training. Price & Tanner (2021) improved the performance of DST by adding additional non-trainable parameters. DST has also demonstrated its strength in feature detection (Atashgahi et al., 2021), lifelong learning (Sokar et al., 2021), federated learning (Zhu & Jin, 2019; Bibikar et al., 2022; Huang et al., 2022), and adversarial training (Özdenizci & Legenstein, 2021). Sparse MoE (Shazeer et al., 2017; Fedus et al., 2021) sparsely activates one of the few expert networks to increase the model capacity – but with a constant computational cost. It usually requires specialized modules, such as gating and selector networks to perform the sparsification.

## 3 FREETICKETS

### 3.1 PRELIMINARIES

**Dynamic Sparse Training.** Let's consider an i.i.d. classification setting with data $\{(x_i, y_i)\}_{i=1}^{N}$, where $x$ usually denotes input samples and $y$ the corresponding labels. For a network $f(\boldsymbol{x}; \boldsymbol{\theta})$ parameterized by $\boldsymbol{\theta} \in \mathbb{R}^d$, we train $f(\boldsymbol{x}; \boldsymbol{\theta})$ to solve the following optimization problem: $\hat{\boldsymbol{\theta}} = \arg\min_{\boldsymbol{\theta}} \sum_{i=1}^{N} \mathcal{L}(f(x_i; \boldsymbol{\theta}), y_i)$.

Dynamic sparse training (DST) starts with a randomly-initialized sparse neural network $f(\boldsymbol{x}; \boldsymbol{\theta}_s)$, parameterized by a fraction of parameters $\boldsymbol{\theta}_s$. The sparsity level of the model is pre-defined as $S = 1 - \frac{\|\boldsymbol{\theta}_s\|_0}{\|\boldsymbol{\theta}\|_0}$, where $\| \cdot \|_0$ is the $\ell_0$-norm. The goal of DST is to yield a sparse network with the target sparsity $S$ after training, while maintaining the overall computational and memory overheads close to training a static sparse model (fixed sparse connectivity).

During training, DST continuously minimizes the loss $\sum_{i=1}^{N} \mathcal{L}(f(x_i; \boldsymbol{\theta}_s), y_i)$, while periodically exploring the parameter space for better sparse connectivity with non-differentiable heuristics. A common exploration heuristics is prune-and-grow, that is, pruning a fraction $p$ of the unimportant weights from $\boldsymbol{\theta}_s$, followed by regrowing the same number of new weights. By repeating this prune-and-grow cycle, DST keeps searching for better sparse connectivities while sticking to a fixed parameter budget. See Appendix A for the general pseudocode and a brief literature review of DST.

## 3.2 FREETICKETS ENSEMBLE

We propose the concept of $FreeTickets$ here. $FreeTickets$ refers to efficient ensemble methods that utilize DST to generate subnetworks for the ensemble. A free ticket is a converged subnetwork created by sparse training methods. These free tickets $\{\theta_s^1, \theta_s^2, \ldots \theta_s^M\}$, observed and collected either within one training run (EDST Ensemble) or multiple training runs (DST Ensemble), are further used to construct the $FreeTickets$ Ensemble. Assuming that the probability of the $k^{th}$ output neuron in the classifier of the $j^{th}$ free ticket is given by $p(a_k^j)$. Then the corresponding output probability in the ensemble is given by taking the average across all the M subnetworks, i.e., $\frac{1}{M} \sum_{j=1}^{M} p(a_k^j)$.

Compared with the existing efficient ensemble techniques (Huang et al., 2017; Wen et al., 2020), $FreeTickets$ induces diversity inspired by the observations that there exist many performant subnetworks with very different sparse topologies located in the full network (Liu et al., 2020b). The efficiency of $FreeTickets$ comes from the fact that each subnetwork is *sparse from the beginning*, so that the memory and floating-point operations (FLOPs) required by $FreeTickets$ can be even fewer than training a single dense network. To realize the concept of $FreeTickets$, we introduce two DST-based ensemble methods, DST Ensemble and EDST Ensemble, as described below.

### 3.2.1 DST ENSEMBLE

Dynamic Sparse Training Ensemble (DST Ensemble) is presented in Algorithm 2, Appendix B. It takes advantage of the training efficiency from DST and independently trains M sparse networks with DST from scratch. By averaging the predictions of each sparse neural network, DST Ensemble can improve the predictive accuracy and uncertainty estimation significantly. Except for the common diversity producers, i.e., random initializations and random stochastic gradient descent (SGD) noise, each DST run converges to different sparse connectivities, promoting even higher diversity over the naive dense Ensemble.

We choose the advanced DST method the Rigged Lottery (RigL) (Evci et al., 2020a) for DST Ensemble. RigL contains three main steps: sparse initialization, model weight optimization, and parameter exploration.

**Sparse Initialization.** Each subnetwork is randomly initialized with the *Erdős-Rényi-Kernel* (ERK) (Mocanu et al., 2018; Evci et al., 2020a) distribution at sparsity of S. The sparsity level of layer $l$ is scaled with $1 - \frac{n^{l-1} + n^l + w^l + h^l}{n^{l-1} \times n^l \times w^l \times h^l}$, where $n^l$ refers to the number of neurons/channels of layer $l$; $w^l$ and $h^l$ are the width and the height of the convolutional kernel in layer $l$. ERK allocates higher sparsities to the layers with more parameters.

**Model Weight Optimization.** After initialization, the activated weights are optimized by the standard optimizer SGD with momentum (Sutskever et al., 2013; Polyak, 1964) and the non-activated weights are forced to zero.

**Parameter Exploration.** After every $\Delta T$ iterations of training, we perform parameter exploration once to adjust the sparse connectivity. More concretely, we first prune a fraction $p$ of weights from $\boldsymbol{\theta}_s$ with the smallest magnitude:

$$\boldsymbol{\theta}_s' = \text{TopK}(|\boldsymbol{\theta}_s|, \ 1 - p) \tag{1}$$

where $\text{TopK}(v, k)$ returns the weight tensor retaining the top $k$-proportion of elements from $v$. Immediately after pruning, we grow the same number of new weights back which have the highest gradient magnitude:

$$\boldsymbol{\theta}_s = \boldsymbol{\theta}_s' \ \cup \ \text{TopK}(|\mathbf{g}_{i \notin \boldsymbol{\theta}_s'}|, \ p) \tag{2}$$

where $\mathbf{g}_{i \notin \boldsymbol{\theta}_s'}$ are the gradients of the zero weights. We follow the suggestions in Liu et al. (2021c) and choose $p = 0.5$ and a large update interval $\Delta T = 1000$ for CIFAR-10/100 and $\Delta T = 4000$ for ImageNet to encourage an almost full exploration of all network parameters.

### 3.2.2 EDST Ensemble

While efficient, the number of training runs (complete training phases) required by DST Ensemble increases linearly with the number of subnetworks, leading to an increased device count and resource requirements. To further reduce the training resources and the number of training runs, we propose Efficient Dynamic Sparse Training Ensemble (EDST Ensemble) to yield many diverse subnetworks in one training run. The overall training procedure of EDST Ensemble is summarized in Algorithm 3.

The challenge of producing many free tickets serially in one training run is how to escape the current local basin during the typical DST training. Here, we force the model to escape the current basin by significantly changing a large fraction of the sparse connectivity, like adding significant perturbations to the model topology. The training procedure of EDST Ensemble is one end-to-end training run consisting of one exploration phase followed by $M$ consecutive refinement phases. The $M$ refinement phases are performed sequentially one after another within one training run.

**Exploration phase.** We first train a sparse network with DST using a large learning rate of 0.1 for a time of $t_{ex}$. The goal of this phase is to explore a large range of the parameter space for a potentially good sparse connectivity. Training with a large learning rate allows DST to validly search a larger range of the parameter space, as the newly activated weights receive large updates and become more competitive at the next pruning iteration.

**Refinement phase.** After the exploration phase, we equally split the rest of training time by $M$ to collect $M$ free tickets. At each refinement phase, the current subnetwork is refined from the converged subnetwork in the previous phase (the first subnetwork is refined from the exploration phase) and then trained with learning rates 0.01 followed by 0.001 for a time of $t_{re}$. Once the current subnetwork is converged, we use a large global exploration rate $q = 0.8$ [1] and a larger learning rate of 0.01 to force the converged subnetwork to escape the current basin. We repeat this process several times until the target number of free tickets is reached. The number of subnetworks $M$ that we obtain at the end of training is given by $M = \frac{t_{total} - t_{ex}}{t_{re}}$. See Appendices I and J for the effect of the global exploration rate $q$ and the effect of different regrowth criteria on EDST Ensemble, respectively.

Different from DST Ensemble, the diversity of EDST Ensemble comes from the different sparse subnetworks the model converges to during each refinement phase. The number of training FLOPs required by EDST Ensemble is significantly smaller than training an individual dense network, as DST is efficient and the exploration phase is only performed once for all ensemble learners.

## 4 EXPERIMENTAL RESULTS

In this section, we demonstrate the improved predictive accuracy, robustness, uncertainty estimation, and efficiency achieved by $FreeTickets$. We mainly follow the experimental setting of MIMO (Havasi et al., 2021), shown below.

**Baselines.** We compare our methods against the dense ensemble, and various state-of-the-art efficient ensemble methods in the literature, including MIMO (Havasi et al., 2021), Monte Carlo Dropout (Gal & Ghahramani, 2016), BatchEnsemble (Wen et al., 2020), TreeNet (Lee et al., 2015), Snapshot (Huang et al., 2017), and FGE (Garipov et al., 2018).

Moreover, to highlight the fact that the free tickets are non-trivial to obtain, we further implement three sparse network ensemble methods: Static Sparse Ensemble (naively ensemble $M$ static sparse networks), Lottery Ticket Rewinding Ensemble (LTR Ensemble; Frankle et al. (2020)) (ensemble $M$ winning tickets[2] trained with the same mask but different random seeds), and pruning and fine-tuning (PF Ensembles; Han et al. (2015)). While Static Sparse Ensemble can be diverse and efficient, it does not satisfy the high expressibility property (Evci et al., 2019). LTR Ensemble suffers from low diversity and prohibitive costs. PF Ensemble requires at least the same training FLOPs as the dense ensemble. See Appendix C for their implementation and hyperparameter details.

**Architecture and Dataset.** We evaluate our methods mainly with Wide ResNet28-10 (Zagoruyko & Komodakis, 2016) on CIFAR-10/100 and ResNet-50 (He et al., 2016) on ImageNet.

---

[1]To be distinct with the standard exploration rate in DST, we define $q$ as the global exploration rate.

[2]We rewind the tickets to the weights at 5% epoch as used in Frankle et al. (2020); Chen et al. (2021b).

**Metrics.** To measure the predictive accuracy, robustness, and efficiency, we follow the Uncertainty Baseline[3] and report the overall training FLOPs required to obtain all the subnetworks including forward passes and backward passes. See Appendix D for more details on the metrices used.

Table 1: Wide ResNet28-10/CIFAR10: we mark the best results of one-pass efficient ensemble in bold and the best results of multi-pass efficient ensemble in blue. Results with * are obtained from Havasi et al. (2021).

| Methods | Acc ($\uparrow$) | NLL ($\downarrow$) | ECE ($\downarrow$) | cAcc ($\uparrow$) | cNLL ($\downarrow$) | cECE ($\downarrow$) | # Training FLOPs ($\downarrow$) | # Training runs ($\downarrow$) |
|---|---|---|---|---|---|---|---|---|
| Single Dense Model* | 96.0 | 0.159 | 0.023 | 76.1 | 1.050 | 0.153 | 3.6e17 | 1 |
| Monte Carlo Dropout* | 95.9 | 0.160 | 0.024 | 68.8 | 1.270 | 0.166 | 1.00× | 1 |
| MIMO (M = 3)* | **96.4** | 0.123 | 0.010 | 76.6 | 0.927 | 0.112 | 1.00× | 1 |
| EDST Ensemble (M = 3) (S = 0.8) (Ours) | 96.3 | 0.127 | 0.012 | **77.9** | 0.814 | 0.093 | 0.61× | 1 |
| EDST Ensemble (M = 7) (S = 0.9) (Ours) | 96.1 | **0.122** | **0.008** | 77.2 | **0.803** | **0.081** | **0.57×** | 1 |
| TreeNet (M = 3)* | 95.9 | 0.158 | 0.018 | 75.6 | 0.969 | 0.137 | 1.52× | 1.5 |
| BatchEnsemble (M = 4)* | 96.2 | 0.143 | 0.021 | 77.5 | 1.020 | 0.129 | 1.10× | 4 |
| LTR Ensemble (M = 3) (S = 0.8) | 96.2 | 0.133 | 0.015 | 76.7 | 0.950 | 0.118 | 1.75× | 4 |
| Static Sparse Ensemble (M = 3) (S = 0.8) | 96.0 | 0.133 | 0.014 | 76.2 | 0.920 | 0.098 | **1.01×** | 3 |
| PF Ensemble (M = 3) (S = 0.8) | **96.4** | 0.129 | 0.011 | 78.2 | 0.801 | 0.082 | 3.75× | 6 |
| DST Ensemble (M = 3) (S = 0.8) (Ours) | 96.3 | **0.122** | **0.010** | **78.8** | **0.766** | **0.075** | **1.01×** | 3 |
| Dense Ensemble (M = 4)* | 96.6 | 0.114 | 0.010 | 77.9 | 0.810 | 0.087 | 4.00× | 4 |

Table 2: Wide ResNet28-10/CIFAR100: we mark the best results of one-pass efficient ensemble in bold and the best results of multi-pass efficient ensemble in blue. Results with * are obtained from Havasi et al. (2021).

| Methods | Acc ($\uparrow$) | NLL ($\downarrow$) | ECE ($\downarrow$) | cAcc ($\uparrow$) | cNLL ($\downarrow$) | cECE ($\downarrow$) | # Training FLOPs ($\downarrow$) | # Training runs ($\downarrow$) |
|---|---|---|---|---|---|---|---|---|
| Single Dense Model* | 79.8 | 0.875 | 0.086 | 51.4 | 2.700 | 0.239 | 3.6e17 | 1 |
| Monte Carlo Dropout* | 79.6 | 0.830 | 0.050 | 42.6 | 2.900 | 0.202 | 1.00× | 1 |
| MIMO (M = 3)* | 82.0 | 0.690 | **0.022** | 53.7 | 2.284 | **0.129** | 1.00× | 1 |
| EDST Ensemble (M = 3) (S = 0.8) (Ours) | 82.2 | 0.672 | 0.034 | **54.0** | **2.156** | 0.137 | 0.61× | 1 |
| EDST Ensemble (M = 7) (S = 0.9) (Ours) | **82.6** | **0.653** | 0.036 | 52.7 | 2.410 | 0.170 | **0.57×** | 1 |
| TreeNet (M = 3)* | 80.8 | 0.777 | 0.047 | 53.5 | 2.295 | 0.176 | 1.52× | 1.5 |
| BatchEnsemble (M = 4)* | 81.5 | 0.740 | 0.056 | 54.1 | 2.490 | 0.191 | 1.10× | 4 |
| LTR Ensemble (M = 3) (S = 0.8) | 82.2 | 0.703 | 0.045 | 53.2 | 2.345 | 0.180 | 1.75× | 4 |
| Static Sparse Ensemble (M = 3) (S = 0.8) | 82.4 | 0.691 | 0.035 | 52.5 | 2.468 | 0.167 | **1.01×** | 3 |
| PF Ensemble (M = 3) (S = 0.8) | 83.2 | 0.639 | 0.020 | 54.2 | 2.182 | 0.115 | 3.75× | 6 |
| DST Ensemble (M = 3) (S = 0.8) (Ours) | **83.3** | **0.623** | **0.018** | **55.0** | **2.109** | **0.104** | **1.01×** | 3 |
| Dense Ensemble (M = 4)* | 82.7 | 0.666 | 0.021 | 54.1 | 2.270 | 0.138 | 4.00× | 4 |

Table 3: ResNet50/ImageNet: we mark the best results of one-pass efficient ensemble in bold and the best results of multi-pass efficient ensemble in blue. Results with * are obtained from Havasi et al. (2021).

| Methods | Acc ($\uparrow$) | NLL ($\downarrow$) | ECE ($\downarrow$) | cAcc ($\uparrow$) | cNLL ($\downarrow$) | cECE ($\downarrow$) | aAcc ($\uparrow$) | aNLL ($\downarrow$) | aECE ($\downarrow$) | Training FLOPs ($\downarrow$) | # Training runs ($\downarrow$) |
|---|---|---|---|---|---|---|---|---|---|---|---|
| Single Dense Model* | 76.1 | 0.943 | 0.039 | 40.5 | 3.200 | 0.105 | 0.7 | 8.09 | 0.43 | 4.8e18 | 1 |
| MIMO (M = 2) ($\rho$ = 0.6)* | 77.5 | **0.887** | **0.037** | **43.3** | 3.030 | 0.106 | 1.4 | 7.76 | 0.43 | 1.00× | 1 |
| EDST Ensemble (M = 2) (S = 0.8) (Ours) | 76.9 | 0.974 | 0.060 | 41.3 | 3.074 | **0.055** | 3.7 | 4.90 | 0.37 | **0.48×** | 1 |
| EDST Ensemble (M = 4) (S = 0.8) (Ours) | **77.7** | 0.935 | 0.064 | 42.6 | **2.987** | 0.058 | **4.0** | **4.74** | **0.35** | 0.87× | 1 |
| TreeNet (M = 2)* | 78.1 | **0.852** | **0.017** | 42.4 | 3.052 | 0.073 | – | – | – | 1.33× | 1.5 |
| BatchEnsemble (M = 4)* | 76.7 | 0.944 | 0.050 | 41.8 | 3.180 | 0.110 | – | – | – | **1.10×** | 4 |
| DST Ensemble (M = 2) (S = 0.8) (Ours) | **78.3** | 0.914 | 0.060 | **43.7** | **2.910** | 0.057 | 4.8 | 4.69 | 0.35 | 1.12× | 2 |
| Dense Ensemble (M = 4)* | 77.5 | 0.877 | 0.031 | 42.1 | 2.990 | 0.051 | – | – | – | 4.00× | 4 |

**Results.** The metrics on CIFAR-10/100 and ImageNet are reported in Table 1, Table 2, and Table 3, respectively. For a fair comparison, we mainly set M of our methods the same as the one used in MIMO. See Appendix E for the comparison between Snapshot, FGE, and our methods.

With multiple training runs, DST Ensemble consistently outperforms other efficient ensemble methods, even the dense ensemble on accuracy, robustness, and uncertainty estimation, while using only 1/4 of the training FLOPs compared to the latter. When the number of training runs is limited to 1, EDST Ensemble consistently outperforms the single dense model by a large margin, especially in terms of accuracy, with only 61% training FLOPs. Moreover, we observe that the performance of EDST Ensemble can further be improved by increasing the sparsity level. For instance, with a high sparsity level S = 0.9, EDST Ensemble can collect 7 subnetworks, more than twice as many as S = 0.8. Combining the prediction of these sparser subnetworks boosts the performance of EDST Ensemble towards the dense ensembles with only 57% training FLOPs, beyond the reach of any efficient ensemble methods. More impressively, DST Ensemble achieves the best performance on uncertainty estimation and OoD robustness among various ensemble methods.

As we expected, Static Sparse Ensemble and LTR Ensemble consistently have inferior ensemble performance compared with DST Ensemble. While PF Ensemble achieves comparable performance

---

[3]https://github.com/google/uncertainty-baselines

with DST Ensemble, its costly procedure requires more than triple the FLOPs of DST Ensemble. We argue that LTR is not suitable for $FreeTickets$ since (1) it starts from a dense network, with the costly iterative train-prune-retrain process. Hence it is more expensive than training a dense network, in contrary to our pursuit of efficient training (our method sticks to training sparse networks end-to-end); (2) The series of sparse subnetworks yielded by iterative pruning is not diverse enough, since the latter sparse masks are always pruned from and thus nested in earlier masks (Evci et al., 2020b). Evci et al. (2020b) show that the different runs of winning tickets discovered by LTR are always located in the same basin as the pruning solution. Consequently, the ensemble of LTR would suffer from poor diversity, leading to poor ensemble performance. We observe a very similar pattern in Section 5.1 as well. The diversity and performance of LTR Ensemble are lower than DST-based ensembles, highlighting the importance of dynamic sparsity in sparse network ensembles.

## 5 FREE TICKETS ANALYSIS

### 5.1 DIVERSITY ANALYSIS

According to the ensembling theory (LeCun et al., 2015; Hansen & Salamon, 1990; Ovadia et al., 2019b), more diversity among ensemble learners leads to better predictive accuracy and robustness. We follow the methods used in Fort et al. (2019) and analyze the diversity of the subnetworks collected by our methods in function space.

Concretely, we measure the pairwise diversity of the subnetworks collected by our methods on the test data. The diversity is measured by $\mathcal{D}_d = \mathbb{E}\left[d\left(\mathcal{P}_1(y|x_1,\cdots,x_N), \mathcal{P}_2(y|x_1,\cdots,x_N)\right)\right]$ where $d(\cdot,\cdot)$ is a metric between predictive distributions and $(x,y)$ are the test set. We use two metrics (1) Prediction Disagreement, and (2) Kullback–Leibler divergence.

**Prediction Disagreement.** Prediction Disagreement is defined as the fraction of test data the prediction of models disagree on: $d_{\text{dis}}(\mathcal{P}_1, \mathcal{P}_2) = \frac{1}{N}\sum_{i=1}^{N}(\arg\max_{\hat{y}_i}\mathcal{P}_1(\hat{y}_i) \neq \arg\max_{\hat{y}_i}\mathcal{P}_2(\hat{y}_i))$.

**Kullback–Leibler (KL) Divergence.** KL Divergence (Kullback & Leibler, 1951) is a metric that is widely used to describe the diversity of two ensemble learners (Fort et al., 2019; Havasi et al., 2021), defined as: $d_{\text{KL}}(\mathcal{P}_1, \mathcal{P}_2) = \mathbb{E}_{\mathcal{P}_1}\left[\log \mathcal{P}_1(y) - \log \mathcal{P}_2(y)\right]$.

Table 4: Prediction disagreement and KL divergence among various sparse ensembles (M = 3, S = 0.8).

| | Wide ResNet28-10/CIFAR10 | | | Wide ResNet28-10/CIFAR100 | | |
|---|---|---|---|---|---|---|
| | $d_{\text{dis}}$ (↑) | $d_{\text{KL}}$ (↑) | Acc (↑) | $d_{\text{dis}}$ (↑) | $d_{\text{KL}}$ (↑) | Acc (↑) |
| LTR Ensemble | 0.026 | 0.056 | 96.200 | 0.111 | 0.185 | 82.100 |
| Static Ensemble | 0.031 | 0.079 | 96.000 | 0.156 | 0.401 | 82.400 |
| EDST Ensemble (Ours) | 0.031 | 0.073 | 96.300 | 0.126 | 0.237 | 82.200 |
| PF Ensemble | **0.035** | **0.103** | **96.400** | 0.148 | 0.345 | 83.200 |
| DST Ensemble (Ours) | **0.035** | 0.095 | 96.300 | **0.166** | **0.411** | **83.300** |
| Dense Ensemble | 0.032 | 0.086 | 96.600 | 0.145 | 0.338 | 82.700 |

We report the diversity among various sparse network ensembles in Table 4. We see that DST Ensemble achieves the highest prediction disagreement among various sparse ensembles, especially on CIFAR-100 where its diversity even surpasses the deep ensemble. Even though EDST Ensemble produces all subnetworks with one single run, its diversity approaches other multiple-run methods. The diversity comparison with non-sparse ensemble methods is reported in Figure 2-Right. Again, the subnetworks learned by DST Ensemble are more diverse than the non-sparse methods and the diversity of EDST Ensemble is very close to the diversity of the dense ensemble, higher than the other efficient ensemble methods e.g., TreeNet and BatchEnsemble.

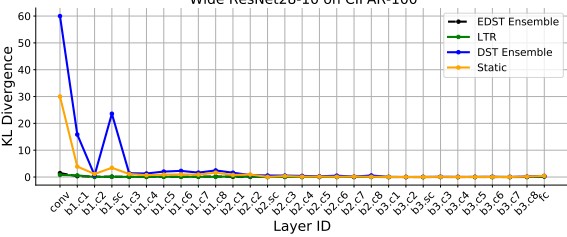

| | $d_{\text{dis}}$ (↑) | $d_{\text{KL}}$ (↑) |
|---|---|---|
| TreeNet | 0.010 | 0.010 |
| BatchEnsemble | 0.014 | 0.020 |
| LTR Ensemble | 0.026 | 0.057 |
| EDST Ensemble (Ours) | 0.031 | 0.073 |
| MIMO | 0.032 | 0.086 |
| DST Ensemble (Ours) | **0.035** | **0.095** |
| Dense Ensemble | 0.032 | 0.086 |

Figure 2: **Left:** KL divergence across intermediate feature maps of various sparse ensemble methods. Each line is the averaged KL divergence among M = 3 subnetworks. **Right:** Diversity comparison between sparse ensemble methods and non-sparse ensemble methods with Wide ResNet28-10 on CIFAR-10 (M = 3, S = 0.8).

To better understand the source of disagreement, we apply KL divergence across intermediate feature maps learned by different subnetworks in Figure 2-Left. Each line is the averaged KL divergence among M = 3 subnetworks. It is interesting to observe that the KL divergence is quite high at early layers for the DST Ensemble and the Static Ensemble, which implies that the source of their high diversity might locate in the early layers. We further provide a disagreement breakdown for the first ten classes in CIFAR-100 between different subnetworks in Appendix G. The overall breakdown of disagreement is very similar to the diversity measurement results, i.e., DST Ensemble > Static Ensemble > EDST Ensemble > LTR Ensemble. We share heatmaps of prediction disagreement of DST/EDST ensemble in Appendix F.

## 5.2 TRAINING TRAJECTORY

In this section, we use t-SNE (Van der Maaten & Hinton, 2008) to visualize the diversity of sparse networks discovered by our methods in function space. We save the checkpoints along the training trajectory of each subnetwork and take the softmax output on the test data of each checkpoint to represent the subnetwork's prediction. t-SNE maps the prediction to the 2D space. From Figure 3 we can observe that subnetworks of DST Ensemble converge to completely different local optima, and the local optima converged by EDST Ensemble are relatively closer to each other. Interestingly, even the last two subnetworks of EDST Ensemble (blue and green lines) start from similar regions, they end up in two different optima.

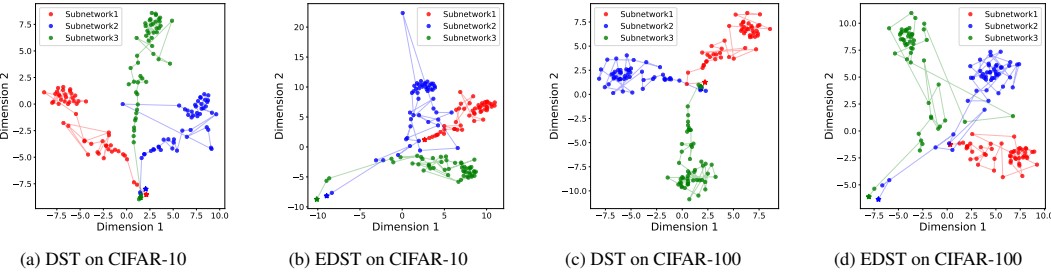

|     (a) DST on CIFAR-10     |     (b) EDST on CIFAR-10     |     (c) DST on CIFAR-100     |     (d) EDST on CIFAR-100     |

Figure 3: t-SNE projection of training trajectories of ensemble learners discovered by DST Ensemble and EDST Ensemble with Wide ResNet28-10 on CIFAR-10/100. The sparsity level is S = 0.8.

## 5.3 ABLATION STUDY OF PARAMETER EXPLORATION

We conduct an ablation study to analyze the effect of parameter exploration on promoting the model diversity. For comparison, we train the same number of subnetworks without any parameter exploration. Shown in Table 5, without parameter exploration, the diversity and the ensemble accuracy consistently decreases, highlighting the effectiveness of parameter exploration on inducing diversity.

Table 5: Ablation study of DST-based Ensemble with and without parameter exploration (M = 3, S = 0.8).

|                                              |    CIFAR10    |    |    |    CIFAR100    |    |    |
| -------------------------------------------- | --------------- | --------------- | --------- | --------------- | --------------- | --------- |
|                                              | $d_{dis}$ (↑) | $d_{KL}$ (↑) | Acc (↑) | $d_{dis}$ (↑) | $d_{KL}$ (↑) | Acc (↑) |
| EDST Ensemble (w/o Parameter Exploration)    | 0.017           | 0.042           | 96.000    | 0.081           | 0.148           | 81.800    |
| EDST Ensemble                                | **0.031**       | **0.073**       | **96.300**| **0.126**       | **0.237**       | **82.200**|
| DST Ensemble (w/o Parameter Exploration)     | 0.031           | 0.079           | 96.000    | 0.156           | 0.401           | 82.400    |
| DST Ensemble                                 | **0.035**       | **0.095**       | **96.300**| **0.166**       | **0.411**       | **83.300**|

## 5.4 EFFECT OF SPARSITY

We further study the effect of subnetworks sparsity on the ensemble performance and diversity, shown in Figure 4. While the individual networks learned by DST Ensemble and EDST Ensemble achieve similar test accuracy, the higher diversity of DST Ensemble boosts its ensemble performance over the EDST Ensemble significantly, highlighting the importance of diversity for ensemble. What's more, it is clear to see that the pattern of the ensemble accuracy is highly correlated with the accuracy of the individual subnetwork. This observation confirms our hypothesis in the introduction, that is, high expressibility is also a crucial desideratum for free tickets to guarantee superior ensemble performance. We further support our hypothesis with the Pearson correlation (Pearson, 1895) and the Spearman correlation (Powers & Xie, 2008) between the accuracy of individual networks and their ensemble accuracy in Appendix H.

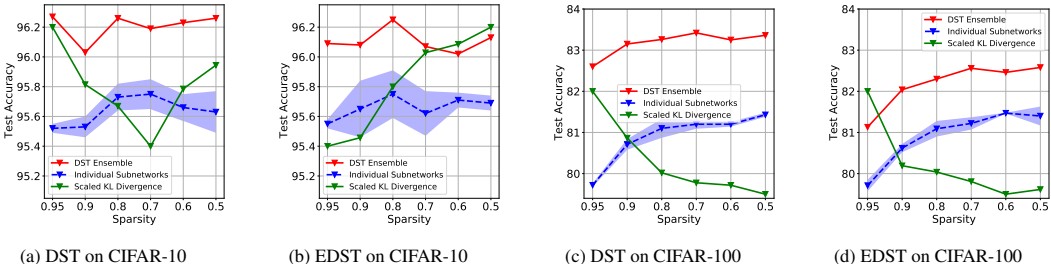

Figure 4: Performance and KL divergence of the subnetworks and the ensemble of subnetworks as sparsity varies. The KL divergence is scaled to the test accuracy for better visualization.

## 5.5 EFFECT OF ENSEMBLE SIZE

In this section, we analyze the effect of the ensemble size (the number of ensemble learners) on our methods. For EDST Ensemble, we fix the total training time and vary the total number of ensemble learners $M$. The larger ensemble size $M$ leads to shorter training time of each ensemble learner. For DST Ensemble, we simply train a different number of individual sparse models from scratch and report the ensemble performance.

The results are shown in Figure 5. "Individual Subnetworks" refers to the averaged accuracy of single subnetworks learned by DST and EDST. The prediction accuracy of DST Ensemble keeps increasing as the ensemble size increases, demonstrating the benefits of ensemble. As expected, the predictive accuracy of individual EDST subnetworks decreases as the ensemble size $M$ increases likely due to the insufficient training time of each subnetwork. Nevertheless, the ensemble performance of EDST is still quite robust to the ensemble size. Specifically, despite the poor individual performance with $M = 7$, the ensemble prediction is still higher than a single dense model. This behavior confirms our hypothesis mentioned before, i.e., multiple free tickets work better than one high-quality winning ticket or even a dense network.

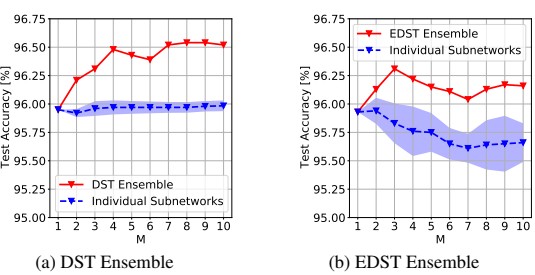

Figure 5: Performance of the subnetworks and the ensemble of subnetworks as the ensemble size $M$ varies (Wide ResNet28-10 on CIFAR-10).

## 5.6 EXPERIMENTS WITH OoD AND ADVERSARIAL ROBUSTNESS

We further test the performance of $FreeTickets$ on out-of-distribution (OoD) detection and adversarial robustness, reporting the results in Appendices K and L, respectively. Our proposed ensemble methods also bring benefits to OoD performance and adversarial robustness.

## 6 CONCLUSION AND FUTURE WORKS

We introduce $FreeTickets$, an efficient way to boost the performance of sparse training over the dense network, even the dense ensemble. $FreeTickets$ is built as a combination of the diverse subnetworks extracted during dynamic sparse training and achieves a co-win in terms of accuracy, robustness, uncertainty estimation, as well as training/inference efficiency. We demonstrate for the first time that DST methods may not match the performance of dense training when standalone, but can surpass the generalization of dense solutions (including the dense ensemble) as an ensemble, while still being more efficient to train than a single dense network.

The aforementioned compelling efficiency has not been fully explored, due to the limited support of the commonly used hardware. Fortunately, some prior works have successfully demonstrated the promising speedup of sparse networks on real mobile processors (Elsen et al., 2020), GPUs (Gale et al., 2020), and CPUs (Liu et al., 2020a). In future work, we are interested to explore sparse ensembles on hardware platforms for real speedups in practice.

## 7 REPRODUCIBILITY

The training configurations and hyperparameters used in this paper are shared in Appendix C. The metrics used to enable comparisons among different methods are given in Appendix D. We have released our source code at `https://github.com/VITA-Group/FreeTickets`.

## 8 ACKNOWLEDGEMENT

We would like to thank Bram Grooten for giving feedback on the camera-ready version; the professional and conscientious reviewers of ICLR for providing valuable comments. Z. Wang is in part supported by the NSF AI Institute for Foundations of Machine Learning (IFML).

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

# A  DYNAMIC SPARSE TRAINING

In this Appendix, we describe the full details of dynamic sparse training. See the survey of Mocanu et al. (2021) and (Hoefler et al., 2021) for more details.

## A.1  ALGORITHM

Dynamic Sparse Training (DST) (Mocanu et al., 2018; Liu et al., 2020a) is a class of methods that enable the end-to-end training of sparse neural networks. DST starts from a sparse network and simultaneously optimizes the sparse connectivity and model weights during training. Without loss of generality, we provide the general pseudocode of DST that can cover most of the existing DST methods in Algorithm 1.

While there is an upsurge in increasingly efficient ways for sparse training (Bellec et al., 2018; Mostafa & Wang, 2019; Dettmers & Zettlemoyer, 2019; Evci et al., 2020a; Liu et al., 2021b; Jayakumar et al., 2020; Liu et al., 2021c; Dietrich et al., 2021; Chen et al., 2021a), most of DST methods contain three key components: sparse initialization, model weight optimization, and parameter exploration. We explain them below.

---

**Algorithm 1** Dynamic Sparse Training

---

**Require:** Network $f_\Theta$, dataset $\{x_i, y_i\}_{i=1}^N$, Layer-wise sparsity ratio: $\mathbb{S}$, Update Interval $\Delta \mathrm{T}$, Exploration Rate $p$

1: # Sparse Initialization
2: $f(\boldsymbol{x}; \boldsymbol{\theta}_s) \leftarrow f(\boldsymbol{x}; \boldsymbol{\theta}; \mathbb{S})$
3: **for** each training step $t$ **do**
4:     # Model Weight Optimization
5:     $f(\boldsymbol{x}; \boldsymbol{\theta}_s) \leftarrow \mathrm{SGD}(f(\boldsymbol{x}; \boldsymbol{\theta}_s))$
6:     **if** $(t \bmod \Delta \mathrm{T}) = 0$ **then**
7:         # Parameter Exploration
8:         Pruning $p$ percentage of parameters using Eq. 1
9:         Growing $p$ percentage of parameters using Eq. 2
10:        Update exploration rate $p$
11:    **end if**
12: **end for**

---

### A.1.1  SPARSE INITIALIZATION

**Layer-wise Sparsity Ratio.** Layer-wise sparsity ratio plays an important role for sparse training. Mocanu et al. (2018) first introduced *Erdős-Rényi* (ER) (Erdős & Rényi, 1959) from graph theory to the field of neural networks, achieving better performance than the uniform sparsity ratio. Evci et al. (2020a) further extended ER to CNN and brings significant gains to sparse CNN training with the *Erdős-Rényi-Kernel* (ERK) ratio. Specifically, the sparsity level of layer $l$ is scaled with $1 - \frac{n^{l-1} + n^l + w^l + h^l}{n^{l-1} \times n^l \times w^l \times h^l}$, where $n^l$ refers to the number of neurons/channels of layer $l$; $w^l$ and $h^l$ are the width and the height of layer $l$. Works with weight redistribution (Mostafa & Wang, 2019; Dettmers & Zettlemoyer, 2019; Liu et al., 2021b) start with the uniform layer-wise ratio and dynamically change the layer-wise ratio according to heuristic criteria, ending up with non-uniform layer-wise ratios.

**Weight Initialization.** Weight initialization also affects the performance of sparse training. Unlike the dense network, sparse networks contain a partial of weights to be zero, breaking the appealing properties of dense initialization such as dynamical isometry (Lee et al., 2020), gradient flow (Evci et al., 2020b; Tessera et al., 2021), etc.

### A.1.2  MODEL WEIGHT OPTIMIZATION

After initializing the sparse model, we need to optimize the model weights to minimize the loss function. In general, dynamic sparse training is compatible with the most widely used optimizers, e.g., minibatch stochastic gradient descent (SGD) (Robbins & Monro, 1951), minibatch stochastic

gradient descent with momentum (Sutskever et al., 2013; Polyak, 1964), and Adam (Kingma & Ba, 2014).

However, we need to pay extra attention to average-based optimizers, such as Averaged Stochastic Gradient Descent (ASGD), which is widely used in the language modeling tasks with RNNs and LSTMs. Liu et al. (2021b) points out that ASGD seriously destroys the sparse connectivity optimization since the average operation brings the newly-grown weights immediately to zero, leading to an over-pruning situation. Moreover, Tessera et al. (2021) shows that optimizers that use an exponentially weighted moving average (EWMA) are sensitive to high gradient flow.

### A.1.3 PARAMETER EXPLORATION

Dynamic sparse training outperforms static sparse training mainly due to the parameter exploration (Liu et al., 2021c). One simple yet effective heuristic used here is prune-and-grow. Removing $p$ percentage of the existing weights and regrowing the same number of new weights allows DST to gradual optimize the sparse connectivity in a heuristic way.

**Weight Prune:** Although there exists various pruning criteria in pruning literature, the most common way for DST is the simple magnitude pruning. We also tried other criteria for pruning at initialization e.g., SNIP (Lee et al., 2019), GraSP (Wang et al., 2020), SynFlow (Tanaka et al., 2020). All of them fall short of the accuracy achieved by magnitude pruning.

**Weight Grow:** The most common ways to grow new weights are random-based growth (proposed in SET (Mocanu et al., 2018)) and gradient-based growth (proposed in RigL (Evci et al., 2020a) and SNFS (Dettmers & Zettlemoyer, 2019)). Random-based growth ensures purely sparse forward pass and backward pass, while it very likely needs more steps to find the important connections, especially for the very extreme sparsities. Gradient-based growth detects the weights that reduce the current loss the fastest, whereas it involves the dense gradient calculation and is likely to cause a collapse of the explored parameter space (Liu et al., 2021b;c; Dietrich et al., 2021).

It is worth noting that the choice of when-to-explore is also of importance for DST. The number of training iterations between two parameter explorations is controlled by the update interval $\Delta T$. Since the newly-grown weights are initialized to zero, a large $\Delta T$ is necessary to guarantee that the new weights receive enough weight updates to survive at the next pruning iteration (Liu et al., 2021c).

## B  PSEUDOCODE OF DST ENSEMBLE AND EDST ENSEMBLE

---
**Algorithm 2** DST Ensemble
---
**Require:** network $f_\Theta$, dataset $\{x_i, y_i\}_{i=1}^N$, ensemble size M, sparsity distribution $\mathbb{S}$, update interval $\Delta$T, DST exploration rate $p$, training time of each free ticket $T$.
1: **for** $j = 1$ **to** M **do**
2:     # Sparse Initialization
3:     $f(\boldsymbol{x}; \boldsymbol{\theta}_s^j) \leftarrow f(\boldsymbol{x}; \boldsymbol{\theta}^j; \mathbb{S})$
4:     **for** $t = 1$ **to** $T$ **do**
5:         # Model Weight Optimization
6:         $f(\boldsymbol{x}; \boldsymbol{\theta}_s^j) \leftarrow \text{SGD}(f(\boldsymbol{x}; \boldsymbol{\theta}_s^j))$
7:         **if** $(t \bmod \Delta\text{T}) = 0$ **then**
8:             # DST Parameter Exploration
9:             DST parameter exploration using Eq. 1 & 2 with $p$
10:             Update exploration rate $p$
11:         **end if**
12:     **end for**
13:     Save the converged sparse subnetwork $f(\boldsymbol{x}; \boldsymbol{\theta}_s^j)$
14: **end for**

---
**Algorithm 3** EDST Ensemble
---
**Require:** network $f_\Theta$, dataset $\{x_i, y_i\}_{i=1}^N$, ensemble size M, sparsity distribution: $\mathbb{S}$, update interval $\Delta$T, DST exploration rate $p$, global exploration rate $q$, training time of the exploration phase $t_{ex}$, training time of each refinement phase $t_{re}/$M
1: # Sparse Initialization
2: $f(\boldsymbol{x}; \boldsymbol{\theta}_s) \leftarrow f(\boldsymbol{x}; \boldsymbol{\theta}; \mathbb{S})$
3: # One Exploration Phase
4: **for** $t = 1$ **to** $t_{ex}$ **do**
5:     $f(x; \boldsymbol{\theta}_s) \leftarrow \text{SGD}(f(\boldsymbol{x}; \boldsymbol{\theta}_s))$
6:     **if** $(t \bmod \Delta\text{T}) = 0$ **then**
7:         DST parameter exploration using Eq. 1 & 2 with $p$
8:     **end if**
9: **end for**
10: # M Sequential Refinement Phases
11: **for** $j = 1$ **to** M **do**
12:     **for** $t = 1$ **to** $t_{re}/$M **do**
13:         $f(\boldsymbol{x}; \boldsymbol{\theta}_s^j) \leftarrow \text{SGD}(f(\boldsymbol{x}; \boldsymbol{\theta}_s^j))$
14:         **if** $(t \bmod \Delta\text{T}) = 0$ **then**
15:             DST parameter exploration using Eq. 1 & 2 with $p$
16:         **end if**
17:     **end for**
18:     Save $f(\boldsymbol{x}; \boldsymbol{\theta}_s^j)$ and escape by parameter exploration using Eq. 1 & 2 with $q$
19: **end for**

## C    IMPLEMENTATION AND HYPERPARAMETERS

In this Appendix, we provide the hyperparameters used in Section 4.

We mentioned in the main content of paper that these free tickets (subnetworks) are non-trivial to obtain due to the three key desiderata. To confirm this, we implement and test two sparsity-based efficient ensemble methods, Static Sparse Ensemble and Lottery Ticket Hypothesis (LTR) Ensemble.

**Static Sparse Ensemble**    Static Sparse Ensemble refers to directly training M sparse subnetworks from scratch and ensemble them for test. Even though it naturally satisfies the low training cost and high diversity, its insufficient accuracy leads to worse performance than the proposed DST/EDST Ensemble.

**LTR Ensemble**    Following Evci et al. (2020b), we use gradual magnitude pruning (GMP) (Zhu & Gupta, 2017; Gale et al., 2019) for LTR, which is a well studied and more efficient pruning method than iterative magnitude pruning (IMP). Concretely, we first use gradual magnitude pruning to prune the dense network to the target sparsity. After that, we retrain and rewind the subnetworks to the 5% epoch for M times. The converged subnetworks are further used for ensemble. Note that even GMP is much more efficient than IMP, the overall training FLOPs required by this method is much higher than directly training a single dense model. We expect that using IMP for Ensemble can be more accurate, but also leads to prohibitive training costs, in contrary to our pursuit of efficient ensemble.

**PF Ensemble**    For the pruning baseline, we choose the strong techniques – global magnitude pruning. We first prune three independently pre-trained dense networks (each is trained for 250 epochs) to the target sparsity, and then fine-tune them for another 250 epochs with different random seeds. We believe this setting can provide enough diversity as all the training procedures are independent, with different random seeds.

**Hyperparameters of DST Ensemble**    For a fair comparison, we follow the training configuration of MIMO. For Wide ResNet28-10 on CIFAR, we train the sparse model for 250 epochs with a learning rate of 0.1 decayed by 10. We use a batch size of 128, weight decay of 5e-4. To achieve a good trade-off between ensemble accuracy and sparsity, we set sparsity as $S = 0.8$. Regarding the hyperparameters of DST, we choose a large update interval $\Delta T = 1000$ between two sparse connectivity updates and a constant DST exploration rate $p = 0.5$.

For ResNet-50 on ImageNet, to the best of our knowledge, all the state-of-the-art DST approaches can not match the performance of the dense ResNet-50 on ImageNet within a standard training time. To guarantee sufficient predictive accuracy for the individual ensemble learners, we follow the training setups used in Liu et al. (2021c) and train each sparse ResNet-50 for 200 epochs with a batch size of 64. The learning rate is linearly increased to 0.1 with a warm-up in the first 5 epochs and decreased by a factor of 10 at epochs 60, 120, and 180. Even with a longer training time, it takes much fewer FLOPs to train sparse networks using DST compared to the dense network. We choose $\Delta T = 4000$ and a cosine annealing schedule for the exploration rate with an initial value $p = 0.5$.

**Hyperparameters of EDST Ensemble**    For Wide ResNet28-10 on CIFAR, we train the 80% sparse model for 450 epochs and 90% sparse model for 850 epochs with $t_{ex} = 150, t_{re} = 100$, so that we can get M = 3 subnetworks with sparsity of 0.8 and M = 7 subnetworks with sparsity of 0.9 for ensemble. The numbers of training FLOPs of this two settings are similar due to the different sparsity levels. The DST exploration rate is chosen as $p = 0.5$ which achieves the best performance as shown in (Evci et al., 2020a; Liu et al., 2021c). To encourage a large diversity between ensemble learners, we use a large global exploration rate $q = 0.8$ to force the subnetowrk escape the current basin. We have also tested a larger exploration rate, i.e., $q = 0.9$. The results are similar with $q = 0.8$.

For ResNet-50 on ImageNet, we set $t_{ex} = 30$ and $t_{re} = 70$ so that we can obtain M = 2 subnetworks with 170 epochs and M = 4 subnetworks with 310 epochs, respectively. We set $p = 0.5$ and $q = 0.8$. The rest of hyperparameters are the same as DST Ensemble. Due to the limited resources, we only test the 80% sparsity for ResNet-50. We believe that the ensemble performance can further improve with lower sparsity.

# D    EXPERIMENTAL METRICS

To measure the predictive accuracy and robustness, we follow the Uncertainty Baseline[4] and focus on test accuracy (Acc), negative log-likelihood (NLL), and expected calibration error (ECE) on the i.i.d. test set, corrupted test sets (i.e., CIFAR-10-C and ImageNet-C) (Hendrycks & Dietterich, 2019) which contain 19 natural corruptions such as added blur, compression artifacts, frost effects, etc, as well as on the natural adversarial samples (i.e., ImageNet-A) (Hendrycks et al., 2019). We adopt {cAcc, cNLL, cECE} and {aAcc, aNLL, aECE} to denote the corresponding metrics on corrupted test sets and natural adversarial samples, respectively.

Negative log-likelihood (NLL) is a proper scoring rule and a popular metric for evaluating predictive uncertainty (Quinonero-Candela et al., 2005).

Expected Calibration Error (ECE) (Naeini et al., 2015), is a widely adopted metric to approximate the difference in expectation between confidence and accuracy of machine learning models (i.e., miscalibration). Specifically, it partitions predictions into M equally-spaced bins, and then calculates a weighted average of the accuracy/confidence discrepancy in each of these bins. Larger ECE values represent a worse match between confidence and accuracy.

To compare the computational efficiency, we report the training FLOPs required to obtain the targeted number of subnetworks for all methods and normalize them with the FLOPs required by a dense model. Following RigL[5] (Evci et al., 2020a), the FLOPs are calculated with the total number of multiplications and additions layer by layer. Briefly speaking, with ERK distribution, the training FLOPs of a sparse Wide ResNet28-10 at sparsity $S = 0.8$ and $S = 0.9$ are 33.7% and 16.7% of the dense model, respectively. For sparse ResNet-50 at $S = 0.8$, the required training FLOPs is 42% of the dense model. We omit the additional computational cost for the extra inputs and the extra outputs of MIMO, as it is negligible compared with the whole training FLOPs. Moreover, we suppose the hardware can fully utilize large batch size so that BatchEnsemble incurs almost no additional computational overhead and memory cost.

# E    COMPARISON WITH SNAPSHOT AND FGE

In this Appendix, we compare our methods with the ensemble methods that use cyclical learning rate schedules to discover diverse dense networks, i.e., Snapshot (Huang et al., 2017) and FGE (Garipov et al., 2018). For a relatively fair comparison, we compare our methods with their '1B' versions (Garipov et al., 2018) which have similar training FLOPs with our methods. However, their test FLOPs would be much larger as their ensemble members are dense networks. As shown in Table 6, while achieving relatively similar performance, Snapshot and FGE require significantly more test FLOPs than the DST-based ensemble methods.

Table 6: Comparison between DST-based ensemble methods with Snapshot (Huang et al., 2017) and FGE (Garipov et al., 2018). The experiments are conducted with Wide ResNet28-10 on CIFAR-10/100.

| Methods | Sparsity | # Training FLOPs (↓) | # Test FLOPs (↓) | Acc CIFAR-10 (↑) | Acc CIFAR-100 (↑) |
|---|---|---|---|---|---|
| Snapshot (M = 12) | 0 | 0.80× | 12.00× | 96.27 | 82.10 |
| FGE (M = 5) | 0 | 0.80× | 5.00× | **96.35** | 82.30 |
| EDST Ensemble (M = 7) (Ours) | 0.9 | **0.57×** | 1.17× | 96.10 | 82.60 |
| EDST Ensemble (M = 3) (Ours) | 0.8 | 0.61× | **1.01×** | 96.30 | 82.20 |
| DST Ensemble (M = 3) (Ours) | 0.8 | 1.01× | **1.01×** | 96.30 | **83.30** |

---

[4]https://github.com/google/uncertainty-baselines

[5]More details can be found in the official repository of RigL https://github.com/google-research/rigl/tree/master/rigl/imagenet_resnet/colabs

## F    PREDICTION DISAGREEMENT ON CIFAR-10/100

We compare the prediction disagreement among the naive dense ensemble, DST Ensemble with sparsity $S = 0.8$, EDST Ensemble with sparsity $S = 0.8$ and $S = 0.9$. The comparison results on CIFAR-10 and CIFAR-100 are shown in Figure 6 and Figure 7, respectively. We find that the subnetworks discovered by DST Ensemble are even more diverse than the ones discovered by the traditional dense ensemble, confirming our hypothesis that dynamic sparsity can provide extra diversity in addition to random initializations. While the average functional diversity of the subnetworks discovered by EDST Ensemble is a bit lower compared with DST Ensemble, the diversity is still notable. Moreover, as expected, at a higher sparsity S = 0.9, the prediction diversity of EDST Ensemble slowly rises as the ensemble size M increases, and ends up with a similar diversity as the Dense Ensemble.

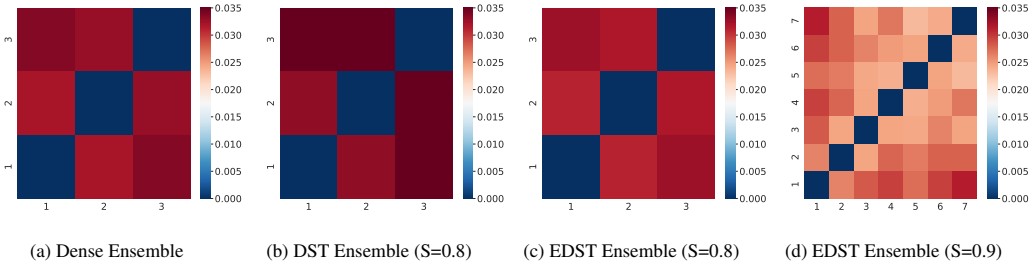

| (a) Dense Ensemble | (b) DST Ensemble (S=0.8) | (c) EDST Ensemble (S=0.8) | (d) EDST Ensemble (S=0.9) |

Figure 6: Prediction disagreement between ensemble learners with Wide ResNet28-10 on CIFAR-10. Each subfigure shows the fraction of labels on which the predictions from different ensemble learners disagree.

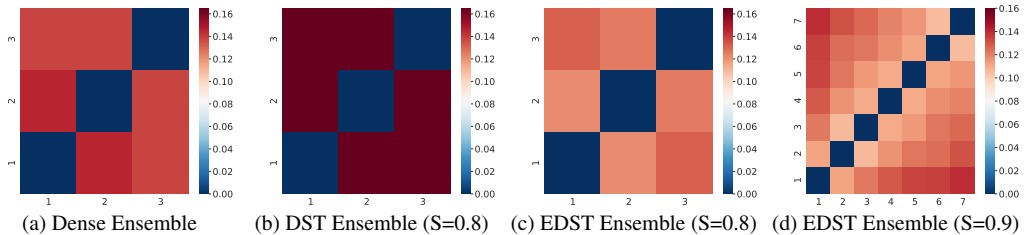

| (a) Dense Ensemble | (b) DST Ensemble (S=0.8) | (c) EDST Ensemble (S=0.8) | (d) EDST Ensemble (S=0.9) |

Figure 7: Prediction disagreement between ensemble learners with Wide ResNet28-10 on CIFAR-100. Each subfigure shows the fraction of labels on which different ensemble learners disagree.

## G  DISAGREEMENT BREAKDOWN BETWEEN VARIOUS SPARSE NETWORK ENSEMBLES ON CIFAR-100

In this Appendix, we try to provide a detailed illustration about various classes that cause the disagreement between various sparse network ensembles. We choose the first 10 classes in CIFAR-100 and measure the disagreement between the subnetworks learned by the same method on these classes. The results are shared in Table 7. The overall breakdown of disagreement is very similar to the diversity measurement results, i.e., DST Ensemble > Static Ensemble > EDST Ensemble > LTR Ensemble. The prediction on classes like Fish, Baby, and Bear have higher diversity than Apple and Beetle.

Table 7: Disagreement Breakdown of various sparse network ensembles, including LTR Ensemble, Static Ensemble, EDST Ensemble, and DST Ensemble, on CIFAR-100.

| **Class** | Apple | Aquarium | Fish | Baby | Bear | Beaver | Bed | Bee | Beetle | Bicycle |
|---|---|---|---|---|---|---|---|---|---|---|
| LTR Ensemble | 0.027 | 0.063 | 0.140 | 0.170 | 0.157 | 0.070 | 0.127 | 0.110 | 0.027 | 0.030 |
| EDST Ensemble | 0.043 | 0.067 | 0.153 | 0.187 | 0.137 | 0.090 | 0.107 | 0.080 | 0.027 | 0.067 |
| Static Ensemble | 0.077 | 0.090 | 0.220 | 0.180 | 0.227 | 0.157 | 0.120 | 0.097 | 0.047 | 0.073 |
| DST Ensemble | 0.057 | 0.080 | 0.227 | 0.277 | 0.230 | 0.137 | 0.160 | 0.133 | 0.053 | 0.093 |

## H  CORRELATION MEASUREMENT BETWEEN THE ACCURACY OF SUBNETWORKS AND THEIR ENSEMBLE ACCURACY

In the main paper, we hypothesize that high expressibility is a crucial desideratum for $FreeTickets$ to guarantee superior ensemble performance. Here, we apply two widely used correlation measurements, Pearson correlation (Pearson, 1895) and Spearman correlation (Powers & Xie, 2008), to support our hypothesis. We use these two methods to measure the correlation between the mean accuracy of individual networks and the accuracy of their ensemble (higher values refer to higher correlation). The results are shared in Table 8. We find that the individual accuracy and the ensemble accuracy are very highly correlated on CIFAR-100. Their correlation is relatively low on CIFAR-10 likely due to the almost saturated accuracy of the individual subnetwork (96%). Given such high accuracy, even the usually clear and strong correlation between the sparsity and test accuracy becomes relatively vague on CIFAR-10 in Figure 4-a and Figure 4-b.

Table 8: Correlation between the mean accuracy of individual networks and the accuracy of their ensemble.

| **Methods** | Wide ResNet28-10/CIFAR10 | | Wide ResNet28-10/CIFAR100 | |
|---|---|---|---|---|
| | Pearson Correlation | Spearman Correlation | Pearson Correlation | Spearman Correlation |
| EDST Ensemble | 0.459 | 0.257 | 0.979 | 0.829 |
| DST Ensemble | 0.332 | 0.319 | 0.972 | 0.812 |

## I EFFECT OF THE GLOBAL EXPLORATION RATE ON EDST ENSEMBLE

Intuitively, a larger global exploration rate $q$ leads to larger sparse connectivity diversity whereas a too large $q$ would prune the important connections, hurting the ensemble accuracy. We perform a sensitivity study with EDST Ensemble in Table 9, whose results are in line with our conjecture. A large global exploration rate $q = 0.8$ promote higher diversity but degrades the ensemble accuracy a bit compared with $q = 0.5$. $q = 0.5$ seems to achieve the best ensemble performance.

Table 9: Effect of Exploration Rate $q$ on EDST Ensemble. "Acc" refers to the ensemble accuracy.

| Different choices of $q$ | Wide ResNet28-10/CIFAR10 | | | Wide ResNet28-10/CIFAR100 | | |
|---|---|---|---|---|---|---|
| | $d_{\text{dis}}$ ($\uparrow$) | $d_{\text{KL}}$ ($\uparrow$) | Acc ($\uparrow$) | $d_{\text{dis}}$ ($\uparrow$) | $d_{\text{KL}}$ ($\uparrow$) | Acc ($\uparrow$) |
| $q = 0.1$ | 0.028 | 0.070 | 96.2 | 0.109 | 0.217 | 82.14 |
| $q = 0.5$ | 0.031 | 0.072 | 96.3 | 0.125 | 2.223 | 82.34 |
| $q = 0.8$ | 0.033 | 0.073 | 96.3 | 0.127 | 0.237 | 82.20 |

## J EFFECT OF REGROWTH METHODS ON EDST ENSEMBLE

Random regrowth naturally considers a larger range of parameters to explore compared to the gradient regrowth as used in the main paper. We added below a small set of experiments to study different growth criteria (random vs gradients) on the EDST Ensemble. As shown in Table 10, it seems that random growth may provide more diversity and better performance for more complex datasets (i.e., CIFAR-100), but more studies are necessary to understand the phenomenon better.

Table 10: Effect of regrowth methods on EDST Ensemble. We compare gradient regrowth and random regrowth. "Individual Acc" refers to the averaged test accuracy of subnetworks discovered by different ensemble methods. "Acc" refers to the ensemble accuracy.

| Methods | Wide ResNet28-10/CIFAR10 | | | | Wide ResNet28-10/CIFAR100 | | | |
|---|---|---|---|---|---|---|---|---|
| | Individual Acc | $d_{\text{dis}}$ ($\uparrow$) | $d_{\text{KL}}$ ($\uparrow$) | Acc ($\uparrow$) | Individual Acc | $d_{\text{dis}}$ ($\uparrow$) | $d_{\text{KL}}$ ($\uparrow$) | Acc ($\uparrow$) |
| Random growth | 95.64 | 0.031 | 0.072 | 96.2 | 81.01 | 0.129 | 0.231 | 82.4 |
| Gradient growth | 95.75 | 0.031 | 0.073 | 96.3 | 81.09 | 0.126 | 0.237 | 82.2 |

# K EXPERIMENTS WITH OOD RESULTS

In this section, we report the performance of OoD detection for single sparse networks and ensemble models. Specifically, we use datasets that are not seen during training time for OoD evaluation. Following the classic routines (Augustin et al., 2020; Meinke & Hein, 2019), SVHN (Netzer et al., 2011), CIFAR-100 (Krizhevsky & Hinton, 2009), and CIFAR-10 with random Gaussian noise (Hein et al., 2019) are adopted for models trained on CIFAR-10 (Krizhevsky & Hinton, 2009); SVHN, CIFAR-10, and CIFAR-100 with random Gaussian noise are used for models trained on CIFAR-100. The ROC-AUC performance (Augustin et al., 2020; Meinke & Hein, 2019) are reported over the respective out of distribution datasets, as shown in Table 11 and Table 12. As we expected, our proposed ensemble methods provide gains to the OoD performance. DST Ensemble and EDST Ensemble can even surpass the dense models by a clear margin in most cases.

Table 11: The ROC-AUC OoD performance for Wide ResNet-28-10 trained on CIFAR-10.

| Methods | OOD dataset for CIFAR-10 trained models | | |
|---|---|---|---|
| | SVHN | CIFAR-100 | Gaussian Noise |
| Static Sparse Ensemble (M = 1) (S = 0.8) | 0.8896 | 0.8939 | 0.9559 |
| Static Sparse Ensemble (M = 3) (S = 0.8) | 0.9229 | 0.9082 | 0.9910 |
| DST Ensemble (M = 1) (S = 0.8) | 0.9082 | 0.8957 | 0.9964 |
| DST Ensemble (M = 3) (S = 0.8) | 0.9533 | 0.9114 | 0.9969 |
| EDST Ensemble (M = 1) (S = 0.8) | 0.9461 | 0.8895 | 0.9607 |
| EDST Ensemble (M = 3) (S = 0.8) | 0.9487 | 0.9045 | 0.9928 |
| EDST Ensemble (M = 1) (S = 0.9) | 0.9439 | 0.8955 | 0.9817 |
| EDST Ensemble (M = 7) (S = 0.9) | 0.9658 | 0.9115 | 0.9956 |
| Single Dense Models | 0.9655 | 0.8847 | 0.9803 |

Table 12: The ROC-AUC OoD performance for Wide ResNet-28-10 trained on CIFAR-100.

| Methods | OOD dataset for CIFAR-100 trained models | | |
|---|---|---|---|
| | SVHN | CIFAR-10 | Gaussian Noise |
| Static Sparse Ensemble (M = 1) (S = 0.8) | 0.7667 | 0.8004 | 0.7858 |
| Static Sparse Ensemble (M = 3) (S = 0.8) | 0.8165 | 0.8141 | 0.9560 |
| DST Ensemble (M = 1) (S = 0.8) | 0.8284 | 0.8019 | 0.8131 |
| DST Ensemble (M = 3) (S = 0.8) | 0.8207 | 0.8221 | 0.8865 |
| EDST Ensemble (M = 1) (S = 0.8) | 0.7481 | 0.7941 | 0.8736 |
| EDST Ensemble (M = 3) (S = 0.8) | 0.8585 | 0.8137 | 0.9046 |
| EDST Ensemble (M = 1) (S = 0.9) | 0.7990 | 0.7937 | 0.8097 |
| EDST Ensemble (M = 7) (S = 0.9) | 0.8092 | 0.8148 | 0.9436 |
| Single Dense Models | 0.7584 | 0.8045 | 0.7374 |

## L    EXPERIMENTS WITH ADVERSARIAL ROBUSTNESS

Table 13 presents the adversarial robustness performance of Wide ResNet-28-10 trained on CIFAR0-10 and CIFAR-100, where the Fast Gradient Sign Method (FGSM) (Szegedy et al., 2013) with step size $\frac{8}{255}$ is adopted. Following a similar routine as in Strauss et al. (2017), we craft adversarial examples for each single model and report the {min, mean, max} robust accuracy over generated attacks from different models. The results demonstrate that our proposed ensemble methods improve robustness accuracy as well, especially in terms of average robust accuracy. More surprisingly, the ensemble of static sparse models can even outperform DST-based Ensembles in this setting. Further analysis on this task serves as interesting future work.

Table 13: The robust accuracy (%) for Wide ResNet-28-10 trained on CIFAR-10 and CIFAR-100.

| Methods | Robust Accuracy (%) {min/mean/max} | |
| --- | --- | --- |
| | CIFAR-10 | CIFAR-100 |
| Static Sparse Ensemble (M = 1) (S = 0.8) | 33.68/38.40/41.29 | 11.89/15.21/17.01 |
| Static Sparse Ensemble (M = 3) (S = 0.8) | 39.28/39.46/39.77 | 16.44/16.81/17.19 |
| DST Ensemble (M = 3) (S = 0.8) | 37.30/38.49/39.12 | 14.96/15.59/15.90 |
| EDST Ensemble (M = 3) (S = 0.8) | 39.66/40.35/41.00 | 13.00/13.19/13.56 |
| EDST Ensemble (M = 7) (S = 0.9) | 35.68/37.74/38.67 | 14.45/14.93/15.56 |
| Single Dense Models | 44.20 | 11.82 |

