# OpenReview forum: "Deep Ensembling with No Overhead for either Training or Testing: The All-Round Blessings of Dynamic Sparsity"
_ICLR.cc/2022/Conference — ICLR 2022 Poster_

### Official Review · Reviewer_hJ1v · 2021-10-29

**Correctness:** 3
**Technical Novelty And Significance:** 2
**Empirical Novelty And Significance:** 3
**Recommendation:** 6
**Confidence:** 4

**Main Review:**

Overall, I find the paper quite timely and exciting. It studies a new dimension where sparse neural networks can be used and has many strengths:
- First large study on the ensemble of sparse neural networks.
- Strong experimental evaluation and results which include Imagenet-r50 ensembles. Especially experiments in ResNet-50 shows significant gains over dense ensembles.
- Useful for the community and has descent potential to inspire further research.

However, the writing and the overall structure of the paper feels a bit rushed: there are many typos and some important baselines/papers are missing. Given the experimental nature of this work, I think it is important to further polish the paper both in terms of writing and experiments. I share my thoughts below:

**1.Major Concerns/Limitations**

a) I think one important ensemble method is missing from the comparison. FGE [2] is possibly the most similar ensemble method to the proposed method. FGE uses cyclic learning rates to find different basins, similar to high q's used in EDST. Comparing these approaches is important I think for completeness. Specifically authors can compare (1) learning rate restart (2) DST training (3) Both LR restart and DST training for EDST.

b) I think this paper has great potential to inspire future research on the topic, therefore I recommend adding pruning baseline to the results. Pruning costs more during training, but might provide better results. It would be nice to see whether that is the case.

c) [3] studies ensembling of sparse NNs and the diversity of various sparse solutions (LT, static, DST). I think a discussion in Section 5.1 comparing results of both papers would be appropriate. Authors can also motivate why LTs are not good for ensembling using their results.

d) Authors say that EDST requires one forward pass, but this is not clear to me; since each model would have a different connectivity and weights and thus would require different (but more efficient than normal) forward pass. The only difference between EDST and DST seems like the training. Do they have different inference procedure? If so, this needs to be clarified; otherwise I think EDST belongs to the same group as DST.

e) Robustness metrics needs to be discussed in the text. I recommend authors to highlight the superior robustness of their approach. It looks like sparse neural networks provide better robustness.

f) An analysis for the p/q used in sparse training would be nice. How does the diversity of the solutions are effected by q?

g) An ablation on importance of grow criteria would be nice (random vs gradients; how much it matter?) An extension of this is to look at the parallel ensemble (i.e. growing different set of connections after removing q=80% of the existing weights); doing this with random would make sense and would tell whether it is better to follow a line, or branch from the same point.

h) There are some things in Figure 4 and 5 that are not clear. In Figure 4, how many networks are ensembled? Given that smaller networks cost less training and inference FLOPs, it feels natural to use more of them for ensembling. Similarly these curves can be all presented on a single plot where x-axis is the inference/training flops and y axis is the ensemble accuracy.

**Minor**
- I am not sure it makes sense to introduce FreeTickets name as the individual methods are introduced later under different synonyms (DST and EDST).
- I found the motivation in the intro a bit off. This paper proposes an efficient ensemble method, however the intro talks about different sparse training methods and their tradeoffs. I think authors can introduce these methods later in the text, while focus on the main problem this paper addresses (making ensemble more efficient?).
- Line 6 and Line 19/20 does the same thing in Algorithm:1, right? It would be nice to use same notation. L20 is currently confusing.
- Escaping current local minima -> escaping current `basin`.
- Equation (2): I think a `union` operator would be more appropriate, "+" is confusing when you combine indices.
- Baselines for the ensemble methods are mentioned in related work briefly. A longer description is needed. It would be nice to explain generic ensemble framework while doing this using the notation introduced (in Preliminaries).
- In different instances authors mention "with can produce in one shot many diverse yet accurate subnetworks", however this is only valid for one of the methods proposed (EDST). I recommend rephrasing them accordingly.
- " Currently, the current only way for DST to match the performance of its dense counterpart on the popular benchmark, e.g., ResNet-50 on ImageNet, is to perform In-Time Over-Parameterization (Evci et al., 2020a; Liu et al., 2021b), " -> do you mean rigl? [1] also achieves this I believe.
- Another related work: Packnet [4].

**Writing**
I think the writing of the paper can be improved significantly. Some of them (stopped at page3):
- "FreeTickets is defined as the ensemble of these sparse subnetworks" -> "Our method, FreeTickets", uses the ensemble of these sparse networks. Overall, I think abstract can be improved. How about starting with the explanation of your method and then having something like: "Since these diverse subnetworks are found through relatively cheap ways we call our method FreeTickets."
- Recent works -> Recent work
- over-extended training time -> longer training time
- ensemble of models, - >ensemble method
- FreeTickets is observed to demonstrate a -> FreeTickets improves over dense baseline in following criteria:
- FreeTickets easily outperforms -> FreeTickets outperforms
- "researchers have investigated efforts to train sparse" ->  researchers studied the training of sparse...
- "promising on training efficiency" -> promising in terms of training.
- "g in general is a tough nut to crack" -> remove i.e "promising in terms of training efficiency, training a sparse network from scr"
- "Currently, the current only way for"
- "We introduce and conduct the concept" -> not sure conducting the concept refers to
- "ds with can p" -> "which can"
- "Our lightest method" -> our light-weight method
- "PRELIMINARY AND SETUPS" -> Preliminaries

[1] https://arxiv.org/abs/2102.07655
[2] https://arxiv.org/pdf/1802.10026.pdf
[3] https://arxiv.org/abs/2010.03533
[4] https://arxiv.org/abs/1711.05769

## After Rebuttal
I read your response and went over the changes made in the main text. I appreciate extra results added during the rebuttal:

It's nice to see sparse connectivity updates complements the learning rate restarts.
DST ensemble matches pruning ensemble and reduces training costs, which is important to highlight.
I believe this paper includes some important and through experimental work on ensemble of sparse networks. I increased my score to 6 (not 8 because it still needs some through polishing; which should be done before the camera-ready if accepted). Below I share some minor comments/suggestions.

- Numbers when comparing random growth with gradient based growth looks suspiciously close. If random works as well as the gradient, it worth demonstrating this clearly, running ResNet-50 experiments might give better signals.
- Algorithm in appendix use p (not q) for exploration.


**Summary Of The Paper:**

To my knowledge this paper provides the first extensive study on the ensemble of sparsely connected neural networks. Ensembles often improve performance, however they are costly. Sparse neural networks, on the other hand, can reduce the cost of each weak learner significantly. In addition to reducing cost at inference, authors also focus on reducing the cost of training, therefore they use end-to-end sparse training methods (dynamic sparse training). Results show that sparse neural networks can provide comparable (or better) ensemble performance while reducing the training and inference cost significantly. This work also provides some interesting insights into the ensemble of sparse neural networks.

**Summary Of The Review:**

The work address an important problem "Making Ensemble training and inference More Efficient" using dynamic sparse training algorithms. Results demonstrate that sparse ensembles (1) reduce training and inference cost and (2) increase the robustness (and sometimes accuracy) over the dense models. Given that sparse models provide same accuracy using less parameters, (1) seems not surprising, however (2) is quite interesting and possibly novel. Authors introduce two algorithms for finding sparse networks however both algorithms seem to do the same thing at inference (contrary to how they are explained); this part and some other points need further clarification. Overall, I believe this work has great potential, however current version lacks few important baselines and written poorly. Therefore my initial rating is weak accept.

---

> ### Author Response · Authors · 2021-11-18
> **Response to Reviewer hJ1v (5/5)**
>
> **5): "Baselines for the ensemble methods are mentioned in related work briefly. A longer description is needed. It would be nice to explain the generic ensemble framework while doing this using the notation introduced (in Preliminaries)."**
> - We will do this in a couple of days.
> **6): "In different instances authors mention "with can produce in one shot many diverse yet accurate subnetworks", however, this is only valid for one of the methods proposed (EDST). I recommend rephrasing them accordingly.**
> - We agree. We have adjusted the phrase to: “our light-weight method (EDST Ensemble) can produce in one shot many diverse yet accurate subnetworks."”
> **7): "Currently, the current only way for DST to match the performance of its dense counterpart on the popular benchmark, e.g., ResNet-50 on ImageNet, is to perform In-Time Over-Parameterization (Evci et al., 2020a; Liu et al., 2021b), " -> do you mean rigl? [1] also achieves this I believe. Another related work: Packnet [4]."**
> - Yes, the In-Time Over-Parameterization (ITOP, introduced in https://arxiv.org/abs/2102.02887) here refers to the longer training time required by the DST methods RigL and SET to match the dense performance. ITOP claims and shows that as long as sufficient parameters have been reliably explored, sparse neural network models trained by Dynamic Sparse Training can usually match and even outperform in a considerable number of cases their dense counterparts. [1] is another interesting paper that uses an additional dense but non-trainable parameter matrix to boost the performance of (dynamic) sparse training.  We will discuss [1] and [4] as related works in the revision.

---

> ### Author Response · Authors · 2021-11-18
> **Response to Reviewer hJ1v (4/5)**
>
> **h): "There are some things in Figure 4 and 5 that are not clear. In Figure 4, how many networks are ensembled? Given that smaller networks cost less training and inference FLOPs, it feels natural to use more of them for ensembling. Similarly these curves can be all presented on a single plot where x-axis is the inference/training flops and y axis is the ensemble accuracy."**
> -  The number of ensemble members in Figure 4 is M=3 for all methods. Since the goal of Figure 4 in Section 5.4 is to demonstrate the effect of sparsity on the ensemble, we fixed the number of subnetworks as M=3 and vary the sparsity of the subnetworks.
>
> - We fully agree that it is preferable to use more of the subnetworks produced by our methods for the ensemble.  We have shown this in Section 5.5 of the original submission. The x-axis of Figure 5 refers to the number of subnetworks used for the ensemble.  The ensemble accuracy of DST Ensemble keeps increasing as the ensemble size increases since each subnetwork is obtained independently.  On the other hand, the ensemble accuracy of EDST Ensemble demonstrates a raising followed by a dropping trend as the ensemble size increases.  For EDST, given a fixed training time, increasing the ensemble size would reduce the refinement time of each subnetwork, leading to performance degeneration. Still, the worst ensemble accuracy in Figure 5b is still higher than one single subnetwork,  indicating the effectiveness of our proposed methods.
>
> **i):  A minor reviewing conflict.**
> - We sincerely thank you for the well-designed, detailed, and constructive comments. After reading your comments very carefully, we find that there is a minor conflict between your rating in the “summary of the review” (i.e., weak accept) and the rating in the “recommendation” (i.e., weak reject). We would highly appreciate if you can fix this minor conflict. We fully respect any decision you make.
>
> ## Minor
>
> **1): "am not sure it makes sense to introduce FreeTickets name as the individual methods are introduced later under different synonyms (DST and EDST)."**
> - Thanks for pointing this out. Our initial goal is to propose the FreeTickets concept which leverages the recently emerged, and very promising, DST to accelerate the costly procedure of deep ensemble.   FreeTickets can cover a wide range of the existing DST methods, such as static sparse training, SET, RIGL, DSR, Top-kast, GraNet. Given the fact that the ML terminology for novel introduced and highly researched topics is prone to be messy and is usually misused, we are trying to use some elaborate concepts to unify the Dynamic Sparse Training (DST) ensembling methods. In our paper, we instantiate and demonstrate the FreeTickets concept within the DST Ensemble and EDST Ensemble using mainly the RigL method. We clarify this aspect in the revised paper.
>
> **2): "I found the motivation in the intro a bit off. This paper proposes an efficient ensemble method, however, the intro talks about different sparse training methods and their tradeoffs. I think authors can introduce these methods later in the text, while focus on the main problem this paper addresses (making ensemble more efficient?)."**
> - Thanks for your suggestion! We agree and will make the suggested change in the revision. In addition to addressing the high-cost problem of the ensemble, we highlight that our proposed paper has also demonstrated for the first time that DST can surpass the generalization of a dense ensemble, while still being more efficient to train than a single dense model. This contribution is recognized and supported by the reviewer HPxD, “This result will drive further research into DST methods, as well as encourage researchers to explore the practicality of DST training for making DNN research more computationally accessible.”
> **3): "Line 6 and Line 19/20 do the same thing in Algorithm:1, right? It would be nice to use same notation. L20 is currently confusing."**
> - Yes, essentially, Line 6 and Line 19/20 do the same thing, i.e., exploring parameters with different exploration rates.  Although essentially similar, their purposes are different. Line 6 is basically the standard parameter exploration (exploration/pruning rate is relatively small) used in all DST methods, whereas Line 19/20 is used to help the subnetwork escape the current basin with a larger pruning rate (q=0.8) in the proposed EDST ensemble method. We use different notations to highlight the difference in case readers might be confused.
> **4): "Escaping current local minima -> escaping current basin; Equation (2): I think a union operator would be more appropriate, "+" is confusing when you combine indices."**
> - Thanks for giving these detailed comments on our writing. We have polished the paper according to your advice.

---

> ### Author Response · Authors · 2021-11-18
> **Response to Reviewer hJ1v (2/5)**
>
> **a): "I think one important ensemble method is missing from the comparison. FGE [2] is possibly the most similar ensemble method to the proposed method... "**
>
> - Thanks for mentioning the important ensemble methods that involve LR restart, e.g., FGE and Snapshot Ensemble (SSE, Huang2017). We are actually aware of the usage of cyclic learning rate used in SSE and FGE and adopt it into the procedure of EDST Ensemble. Each time the current subnetwork converges with the small learning rate of 0.001, we force it to escape the local basin with large q and continually train it with relatively larger learning rates of 0.01 followed by 0.001 until it converges to another basin. Our ablation study in Figure 5 has shown both LR restart and dynamic sparsity contribute significantly to the high performance. In this case, the ’’EDST Ensemble (w/o Parameter Exploration)” refers to the sparse version of Snapshot, whose performance and diversity fall short of the EDST Ensemble. For convenience, we also share the results here.
> | |  | WRN-28-10 CIFAR-10 | |  | WRN-28-10 CIFAR-100 | |
> |:------------- |:------------- |:-----------: |:-----------:| :-----------: |:-----------:|:-----------:|
> |                   | $d_{dis}$ | $d_{KL}$ | Acc|$d_{dis}$ | $d_{KL}$ | Acc|
> |EDST Ensemble w/o Explo (LR restart) |  0.017 | 0.042 | 96.0 | 0.081 | 0.148 | 81.8|
> |EDST Ensemble (DST+LR restart) |  0.031 | 0.073 | 96.3 | 0.126 | 0.237 | 82.2 |
> |EST Ensemble (DST) |  0.035 | 0.095 | 96.3 | 0.166 | 0.411 | 83.3 |
> - Moreover, we also compare our methods with FGE and SSE below. "PF Ensemble (M=3)" refers to pruning and fine-tuning Ensemble. For a relatively fair comparison, we compare our methods with their ‘1B’ versions which have similar training FLOPs with our methods. However, their test FLOPs would be much higher than our methods as their ensemble members are dense networks. We can see that our methods require to have much fewer test FLOPs compared with FGE and SEE.
> | |  |  |  | WRN-28-10 CIFAR-10 | WRN-28-10 CIFAR-100  |
> |:------------- |:------------- |:-----------: |:-----------:| :-----------: |:-----------:|
> | Methods | Sparsity | Training FLOPs | Test FLOPs | Acc | Acc|
> | Dense Ensemble (M=4) | 0 | 4x | 4x | 96.6 | 82.7 |
> | SSE (M=12)| 0 | 0.8x | 12x | 96.27 | 82.1|
> | FGE (M=5)| 0 | 0.8x | 5x | 96.35 | 82.3|
> | PF Ensemble (M=3) | 0.8 | 3.75x | 0.75x | 96.4 | 83.2|
> | EDST Ensemble (M=7) | 0.9 | 0.57x | 1.17x | 96.1 | 82.6|
> | EDST Ensemble (M=3) | 0.8 | 0.61x | 1.01x | 96.3 | 82.2|
> | DST Ensemble (M=3) | 0.8 | 1.01x | 1.01x | 96.3 | 83.3|
>
> **b): "I think this paper has great potential to inspire future research on the topic, therefore I recommend adding pruning baseline to the results. Pruning costs more during training, but might provide better results. It would be nice to see whether that is the case."**
> - Thank you for your suggestion! We fully agree with you and have added the pruning and fine-tuning (PF Ensemble) baseline (see table in the above answer). We use the strong global magnitude pruning and prune and finetune three independently pre-trained dense networks (trained with 250 epochs). We believe this setting can provide enough diversity as all the training procedures are independent, with different random seeds. As you expected, pruning costs more during training. Impressively, our DST Ensemble achieves equally good performance compared with pruning and fine-tuning Ensemble with only ⅓ training FLOPs and much fewer memory requirements. This highlights the great potentials of DST training for making DNN research more computationally accessible.

---

> ### Author Response · Authors · 2021-11-18
> **Response to Reviewer hJ1v (1/5)**
>
> We would like to thank the reviewer for recognizing the novelty and importance of our study on the ensemble of sparsely connected neural networks. Your knowledgeable and detailed comments have significantly made our paper stronger. Below we clarify your questions and we sincerely hope they could address all your concerns:
>
> ## Major
>
> **d) & Differences between DST and EDST Ensemble: "Authors introduce two algorithms for finding sparse networks however, both algorithms seem to do the same thing at inference (contrary to how they are explained). Authors say that EDST requires one forward pass, but this is not clear to me; The only difference between EDST and DST seems like the training. Do they have different inference procedure?."**
>
> - We believe there is some confusion/misunderstanding regarding the procedures of the DST Ensemble and EDST Ensemble, which is caused by the problematic usage of “forward passes” in section 3.3. By one forward pass, we mean EDST Ensemble only requires one training run to obtain all the ensemble members. As we mentioned in the introduction “EDST can yield many free tickets in one single run.”, the advantage of EDST is its one training run (not in the inference procedure). We apologize for this confusion and will upload the fixed revision in a couple of days.
>
> - Yes, you are correct that both algorithms simply average the prediction for the ensemble at inference. The main difference between DST Ensemble and EDST Ensemble consists in the training procedure. The training procedure of DST is a straightforward ensemble of M DST models trained independently. On the other hand, the training procedure of EDST Ensemble is technically one single training run consisting of 1 Exploration Phase followed by M consecutive/sequential Refinement Phases.
>
> - For EDST, please note that we do not refine the original subnetwork obtained in the Exploration Phase M times. Instead, the M Refinement Phases are performed sequentially one after another within one run. We only train one single subnetwork with dynamic connectivity during the whole course of EDST training. At any moment in time, the current subnetwork is refined from the converged subnetwork in the previous Refinement Phase. The procedure of EDST is similar to the Snapshot Ensemble. Once the current subnetwork is converged, we use a large exploration rate q and larger learning rates to force the converged subnetwork to escape the current basin. The converged subnetwork during each refinement phase is saved for the test ensemble. We will add more elaborate descriptions in the revision. Compared with the traditional dense ensemble which requires training multiple networks independently, these procedures corroborated with the fact that we are using sparse neural networks make our proposed methods much more efficient.

---

### Official Review · Reviewer_HPxD · 2021-11-02

**Correctness:** 4
**Technical Novelty And Significance:** 2
**Empirical Novelty And Significance:** 3
**Recommendation:** 6
**Confidence:** 4

**Main Review:**

Strengths:
- Well written paper, with good background, clear motivation.
- This paper's contributions, while completely empirical, present a very good and I'd say relatively complete experimental analysis that will no doubt be useful for researchers and engineers in understanding how DST can benefit sparse neural network training in generalization and/or efficiency.
- Results for EDST are compelling when generalization/theoretical training compute is considered (as it should be), results that I believe are novel and important - showing that dynamic sparse training can match dense generalization with less (theoretical compute).


Weaknesses:
- DST ensemble is not novel, as it is just an ensemble of DST-trained models which has been analyzed (although with a different focus) in other work.
- EDST is marginally more novel, but clearly related to other existing methods of training deep neural networks.
- Some of the analysis of the diversity of sparse solutions is not novel, e.g. Lottery Ticket rewinding not resulting in diverse solutions/producing poor ensembles relative to static sparse ensembles is one of the main contributions of the paper "Gradient flow in sparse neural networks", Evci et al., 2020. This paper is cited by the authors elsewhere but not in reference to these results. The authors expand on these results however, and I believe it would strengthen the author's results on the diversity of solutions if they cited this work in the appropriate places and compared their findings (which I believe corroborate that paper's).
- Some parts of the method are not clearly described (i) through (iv) below:
(i) 3.1, end of page 3: "A free ticket is one of the sparse neural networks (subnetworks) generated *during* the dynamic sparse training procedure". These free tickets \theta_s^1, \theta_s^2, \ldots, \theta_s^M)..." this is *very* misleading as "during" the DST training there are multiple subnetworks generated over time with each mask update, and this text makes it sounds like the ensemble is of those - but I realized by the next page this is not the case. This needs rewording to make it clear subnetworks are the result of multiple DST training runs.
(ii) example Algorithm 1 does not describe how the refinement model weights (\theta_s^j) are initialized explicitly, and this isn't clearly defined in the text. I can only reasonably assume they are initialized from \theta_s in the exploration phase.
(iii) Another aspect of this method that's not clear: "...at the end of each refinement phase" in section 3.3, does this mean the refinement phase is run multiple times? Or is this referring to each of the subnetwork's refinements? Figure 1 also confuses me on this point as it seems to show multiple refine/exploration steps, and Algorithm 1 only seems to show one Refinement phase (of t_{ex} < ?? steps). Given the lack of clarity, I've assumed there is only one exploration & M refinement phases (consisting of M subnetwork training runs) but I would appreciate this being made clear in the text/Algo 1/Fig 1.
(iv) several of the variables used in Algo 1 are undefined or misused, including s^i \in (s^1, \ldots, s^L) which seems to be the sparse masks, and t_{ex}, and t_{re} which are used as time counters/indices in the algorithm, but are the total amount of time in the text.
- The paper's usage of "forward passes" is problematic - it's defined as a "complete training phase" in 3.3, i.e.  a whole training run, which goes against the common usage of the term and is very misleading for the careless reader in the results - change this term to "training runs"
- Although the motivation is efficiency, there are no real timings *or* explanation on how these theoretical FLOPS gains might be realized. There are compelling papers that the author could refer to showing how to accelerate unstructure sparsity however on CPU and GPU (see "Fast sparse convnets", Gale et al. 2020), and I would encourage the authors to refer to these to allay such concerns.
- Figure 1 is much too small, and yet there is a lot of wasted space in how it's presented on the page. I'm also pretty sure that ICLR doesn't allow text to wrap around figures like this - but I could be wrong. Regardless, it would be very easy to fix this figure so that it was more compact, a proper floating figure, and more legible at the same time.

**Summary Of The Paper:**

The authors propose to improve the generalization of dynamic sparse neural network training methods, such as RiGL, by learning an ensemble of these models. The authors propose two methods of doing so, the Dynamic Sparse Training (DST) Ensemble, and the Efficient Dynamic Sparse Training (EDST) Ensemble. A DST ensemble is a straightforward ensemble of M DST trained models, which like all ensembles incurs high overhead in training and inference cost (training/using M models). An EDST ensemble on the other hand uses approximately the same training compute as a single model. It does so by splitting the training into two phases (of the total amount of steps as a single standard DST training run). The first phase is standard DST training (called the Exploration Phase), where training uses a large learning rate. The second phase (called the Refinement Phase) trains M models initialized from the weights of the model in the first phase, but with much smaller learning rates decayed over training. Hyperparameters determine the amount of exploration v.s. refinement training steps.

The authors evaluate the trained ensembles for Image Classification with Wide ResNet28-10 on CIFAR-10/100 and ResNet-50 on ImageNet,  and compare these to baselines including: an ensemble of sparse trained models (with a static mask), an ensemble of Lottery Ticket Rewound models, an alternative method with similar motivation (MIMO), a single dense model, BatchEnsemble and TreeNet. The DST and EDST Ensembles show similar or marginally higher generalization than baselines, but specifically in the EDST case, with much fewer (theoretical) training FLOPS. The authors also compare the diversity of the ensemble members in a few of these baselines, showing that DST has much higher diversity than EDST (as might be expected), but also interestingly that DST appears to have higher diversity than a dense ensemble. The authors perform an ablation study on their results, attempting to discern the affects of parameter exploration (i.e. dynamic mask updates in DST methods), affect of sparsity and affect of ensemble size on the results. The authors also claim to show better out-of-distribution performance and robustness, although these results are in the appendix and only referred to in the main text.

**Summary Of The Review:**

I believe this paper, albeit short on technical novelty, does provide a sufficiently novel and important empirical evaluation and details analysis of Dynamic Sparse Training (DST) ensembling methods, and could be accepted - assuming the authors can address the issues I have detailed in my review. In particular this paper has demonstrated, I believe for the first time, that DST methods such as RiGL may not match dense training when standalone, but can surpass the generalization of a dense solution as an ensemble - while still being (theoretically) more efficient to train than the dense model. This result will drive further research into DST methods, as well as encourage researchers to explore the practicality of DST training for making DNN research more computationally accessible.

---

> ### Author Response · Authors · 2021-11-18
> **Response to Reviewer HPxD  (2/2)**
>
> **Q4: "Some parts of the method are not clearly described (i) through (iv) below:**
>  - **(i) 3.1, end of page 3: "A free ticket is one of the sparse neural networks (subnetworks) generated during the dynamic sparse training procedure". These free tickets \theta_s^1, \theta_s^2, \ldots, \theta_s^M)..."**
> Thanks for your comment. This sentence indeed can be misleading. We will replace it with a more detailed description, like “For DST Ensemble, a free ticket is the final converged subnetwork of each DST training run; For EDST Ensemble, a free ticket is a subnetwork that converges at the end of each Refinement Phase.”
>  - **(ii) example Algorithm 1 does not describe how the refinement model weights (\theta_s^j) are initialized explicitly, and this isn't clearly defined in the text. I can only reasonably assume they are initialized from \theta_s in the exploration phase.**
>  - **(iii) Another aspect of this method that's not clear: "...at the end of each refinement phase" in section 3.3, does this mean the refinement phase is run multiple times? Or is this referring to each of the subnetwork's refinements? Figure 1 also confuses me on this point as it seems to show multiple refine/exploration steps, and Algorithm 1 only seems to show one Refinement phase (of t_{ex} < ?? steps). Given the lack of clarity, I've assumed there is only one exploration & M refinement phases (consisting of M subnetwork training runs) but I would appreciate this being made clear in the text/Algo 1/Fig 1.**
> We answer these two questions together. For EDST Ensemble in Algorithm 1 or Fig 1, we indeed have M Refinement Phases but they are performed one after another sequentially within one training run (not independently with M runs). Once the current refinement phase ends, we save the converged subnetwork and use a large exploration rate q to escape the current basin and start the next Refinement Phase. Thus, the whole procedure can be represented in Figure 1, where one initial Exploration phase and M refinement phases are performed sequentially within one training run. We obtain one subnetwork at the end of each Refinement phase. We will make this clearer in the text/Algo 1/Fig 1.
>  - **(iv) several of the variables used in Algo 1 are undefined or misused, including s^i \in (s^1, \ldots, s^L) which seems to be the sparse masks**
> s^i \in (s^1, \ldots, s^L) refers to the sparsity of layer from layer i to L.
> **Q5: The paper's usage of "forward passes" is problematic - it's defined as a "complete training phase" in 3.3, i.e. a whole training run, which goes against the common usage of the term and is very misleading for the careless reader in the results - change this term to "training runs"**
> - We highly appreciate your careful reading and kind reminder! We will change ‘forward pass’ to ‘training runs’ in the revision.
>
>  **Q6: Although the motivation is efficiency, there are no real timings or explanation on how these theoretical FLOPS gains might be realized. There are compelling papers that the author could refer to showing how to accelerate unstructure sparsity however on CPU and GPU (see "Fast sparse convnets", Gale et al. 2020), and I would encourage the authors to refer to these to allay such concerns.**
>
> - We fully agree with you on this point. Despite the promising performance and efficiency demonstrated by sparse neural networks, the theoretical reduction of FLOPs is difficult to translate into real-world gains owing to the limited support of sparse operations on the commonly used hardware. Also due to this reason, we chose the theoretical training/inference FLOPs to enable comparisons among various methods, the same as most of the papers on the topic. Works like  "Fast sparse convnets (Elsen et al. 2020)" and “Sparse GPU kernels for deep learning” (Gale et al. 2020) that achieve practical speedups on GPUs and CPUs are highly appreciated and will significantly help the researchers in the sparsity community. We will definitely add a paragraph in the revision to introduce these works and to discuss how they can help in the future to efficiently implement the proposed ensemble methods.
>
>  **Q7: Figure 1 is much too small, and yet there is a lot of wasted space in how it's presented on the page. I'm also pretty sure that ICLR doesn't allow text to wrap around figures like this - but I could be wrong. Regardless, it would be very easy to fix this figure so that it was more compact, a proper floating figure, and more legible at the same time.**
>  - Thanks for your suggestion! We will adjust it in the revision!

---

> > ### Comment · Reviewer_HPxD · 2021-11-23
> > **Thanks for the clarifications/answers.**
> >
> > I'd just like to acknowledge the author's comments/response to my questions which I believe cleared up much of the confusion I had as to the algorithm as presented in the text/figures.

---

> ### Author Response · Authors · 2021-11-18
> **Response to Reviewer HPxD (1/2)**
>
> We sincerely appreciate your positive comments and support! We also believe in the great potential of DST training for making the increasingly costly DNN research more computationally accessible and stronger. We address your detailed comments below.
>
>  &nbsp;
>  &nbsp;
>
> **Q1: "DST ensemble is not novel, as it is just an ensemble of DST-trained models which has been analyzed (although with a different focus) in other work."**
>
> - We believe that this is a fair point. We are grateful and encouraged by your recognition of our novelty in the evaluation and analysis of the Dynamic Sparse Training (DST) ensemble. While this paper does not present a novel algorithm in the original dynamic sparse training sense for single networks, it does presents novel insight, experiments, and analysis of DST in terms of a totally different perspective, i.e., deep ensemble, all of which are outside of the scope of the DST's typical papers. With extensive empirical studies, our paper provides insights into two important questions. (1) How to boost the performance of DST methods over the dense training, while maintaining their superior training efficiency? (2) How to accelerate the powerful but costly deep ensemble?
>
> **Q2:  "EDST is marginally more novel, but clearly related to other existing methods of training deep neural networks."**
> - Up to our knowledge, in the sparse training work, EDST is the first of its kind as detailed further. In the second phase of EDST (Refinement Phase), not all M subnetworks are initialized from the weights of the model obtained in the first phase (Exploration Phase). The training procedure of EDST Ensemble is technically one single training run consisting of 1 Exploration Phase followed by M consecutive/sequential Refinement Phases. We do not refine the original subnetwork obtained in the Exploration Phase M times independently. Instead, the M Refinement Phases are performed sequentially one after another within one run. At a moment in time, the current subnetwork is refined from the converged subnetwork in the previous Refinement Phase. Somehow, the procedure of EDST is similar to the Snapshot Ensemble (https://arxiv.org/abs/1704.00109) which focuses on dense neural networks. Once the current subnetwork is converged, we save it for test and use a large exploration rate q to escape the current basin. Having diversity also in the connectivity pattern itself from sparse training, and not just in the weights values (from dense training), ensures at the end better performance as it is reflected by Tables 1,2,3 and answer a) to the Reviewer **hJ1v**.
>
> **Q3: "Some of the analysis of the diversity of sparse solutions is not novel, e.g. Lottery Ticket rewinding not resulting in diverse solutions/producing poor ensembles relative to static sparse ensembles is one of the main contributions of the paper "Gradient flow in sparse neural networks", Evci et al., 2020."**
> - Thanks for your suggestion. We fully agree with you that the contribution of LTR does not result in diverse solutions is one of the main contributions from Evci et al., 2020. Our results on LTR in the submission were used to have a baseline and to conduct a comparison between different sparse ensemble methods. Indeed,  Evci et al., 2020 is an excellent paper with a different goal in mind. The motivation of the ensemble study in Evci et al., 2020 is to understand LTs. They conduct extensive and detailed experiments to show that the different tickets discovered by LTs are always located in the same basin as the pruning solution. Consequently, the ensemble of LTs would suffer from poor diversity, leading to poor ensemble performance due to the lack of diversity. We also observe a very similar pattern in our paper. The diversity and ensemble performance of LTR is lower than DST-based ensembles, evidentiating the role of dynamic sparsity in promoting diversity. As suggested by the reviewer, in the revised version, we will use the results from Evci et al., 2020, to strengthen our claims.

---

### Official Review · Reviewer_4rju · 2021-11-02

**Correctness:** 3
**Technical Novelty And Significance:** 4
**Empirical Novelty And Significance:** 3
**Recommendation:** 6
**Confidence:** 4

**Main Review:**

Strengths
- The outline of the training setups across different versions of DST and the proposed method was extremely clear and direct. It helps the reader understand the distinction between them easily.
- Conceptually and practically, the improvements offered by training an ensemble of sparse networks to augment the failures of a single dense/sparse model is extremely useful. A stronger emphasis on analyzing the structural and representational diversity among the FreeTickets and drawing comparisons to prior literature on cascaded weak classifiers would help deepen the impact of the work.

Weaknesses
- The term "cheap" pertaining to one of the key properties required of FreeTickets is left undefined until Pg. 4, Section 3.1. While the comparison to dense networks and their ensembles is fair from the perspective of performance, could the authors clarify if methods like static LTR ensemble and others are similarly "cheap" to  DST and EDST? Ultimately, tying a quantifiable comparison of total FLOPs to calculate ensembles to emphasize the notion cheap would be preferable.
- Could the authors clarify what $n^{(L)}$ refers to in Pg. 4, L. 2?
- Could the authors discuss the different decision factors that influence the choice of $t_{ex}$? While M controls the amount of time each sub-network is fine-tuned, to what degree does $t_{ex}$ need to be explored so that a sufficient embedding space is learned?
- Pg. 6, Paragraph on Baselines, the last sentence mentions that the LTR ensemble has low diversity and prohibitive costs. A quantifiable measure or citation would help qualify this statement.
- Could the authors clarify if the training FLOPs include the total number of forward passes and the FLOPs for each pass or they quantify the FLOPs required for a single pass only?
- The difference in ECE values between the best and remaining methods across different models and datasets is unstable. Could the authors comment on the significance of these values and their changes? These become critical as we observe a large variation in the best value across multiple methods and metrics, considering Tables 1, 2 and 3.
- A breakdown of the various labels/samples that cause the disagreement between various FreeTickets would be instructional and add a visual cue to the diversity metric. Additionally, applying KL divergence across intermediate feature embeddings could trace the source of variations and how they have an effect on various metrics.
- Fig. 3 and Section 5.2 do not comprehensively cover the tSNE projections of other models tested in Tables 1,2 and 3. Without a clear frame of reference and common scale factors it is difficult to compare across the various projections presented. I encourage the authors to try providing a projections for multiple baseline and remove any possible stochasticity in projections to ensure the comparisons are suitable.
- Section 5.4: Based on Fig. 4 there is a claim that the pattern of ensemble accuracy is highly correlated with the accuracy of the individual sub-network. I encourage the authors to support this with a quantifiable measure. Additionally, visually they do  not seem to have the same pattern of evolution, wrt. sub-network accuracy or deviation in performance of sub-networks. Could the authors clarify the statement in the manuscript?
- Section 5.5: The manuscript uses the term "Individual Tickets" which does not match the nomenclature used in Fig. 5.
- Section 5.6: The adversarial robustness results for CIFAR-10 show that the dense model is more robust than the sparse models. Could the authors clarify their statements in Section 5.6?

After Rebuttal

I thank the authors for their detailed response to each of the comments posted and a well done revision of their manuscript.
Based on their responses, I have updated my recommendation to an acceptance.

For the comment on t-SNE plots, on of the key issues when comparing across vastly differing feature spaces is that the 2D projection scales unequally. In providing such projections for alternative ensembling approaches, we can visually discuss the diversity of solutions (mirroring the discussion based on quantifiable metrics).


**Summary Of The Paper:**

This work focuses on generating ensembles of sparse networks, called FreeTickets, that requires significantly lesser training FLOPs and parameters compared to single or ensemble dense networks while maintaining high prediction accuracy.  FreeTickets are expected to be diverse, highly expressive and easy to obtain. Apart from the methodology to generate these FreeTickets and ensure they match the desired properties, a key claim in this work is that FreeTickets demonstrate improved prediction accuracy, robustness and uncertainty estimation while being much more efficient, in terms of the computational resources required for training.

**Summary Of The Review:**

The proposed methodology is supported strongly by the main set of results. However, there are some missing explanations and justifications required for statements in the analysis performed that need to be clarified.

---

> ### Author Response · Authors · 2021-11-18
> **Response to Reviewer 4rju (4/4)**
>
> [1] [Evci, Utku, et al. "Gradient flow in sparse neural networks and how lottery tickets win." arXiv preprint arXiv:2010.03533 (2020).](https://arxiv.org/abs/2010.03533)
>
> [2] [Naeini, Mahdi Pakdaman, Gregory Cooper, and Milos Hauskrecht. "Obtaining well calibrated probabilities using bayesian binning." Twenty-Ninth AAAI Conference on Artificial Intelligence. 2015.](https://people.cs.pitt.edu/~milos/research/AAAI_Calibration.pdf)
>
> [3] [Shiwei Liu, Lu Yin, Decebal Constantin Mocanu, and Mykola Pechenizkiy. Do we actually need dense over-parameterization? in-time over-parameterization in sparse training. ICML 2021.](https://arxiv.org/abs/2102.02887)
>
> [4] [Guo, Chuan, et al. "On calibration of modern neural networks." ICML, 2017](https://arxiv.org/abs/1706.04599)
>
> [5] [Havasi, Marton, et al. "Training independent subnetworks for robust prediction." ICLR 2021.](https://openreview.net/forum?id=OGg9XnKxFAH)
>
> [6] [Fort, Stanislav, Huiyi Hu, and Balaji Lakshminarayanan. "Deep ensembles: A loss landscape perspective." arXiv preprint arXiv:1912.02757 (2019).](https://arxiv.org/abs/1912.02757)

---

> ### Author Response · Authors · 2021-11-18
> **Response to Reviewer 4rju (3/4)**
>
>
> **Q8: "Fig. 3 and Section 5.2 do not comprehensively cover the tSNE projections of other models tested in Tables 1,2 and 3. Without a clear frame of reference and common scale factors it is difficult to compare across the various projections presented. I encourage the authors to try providing a projections for multiple baseline and remove any possible stochasticity in projections to ensure the comparisons are suitable."**
>
> - The main goal of tSNE projections in Fig. 3 is not to conduct a comparison among different efficient ensemble methods. Instead, we use the tSNE tool to visualize the training trajectory of different ensemble members (subnetworks) discovered by the same method. This experiment is important as one of our methods EDST obtains many subnetworks within one single run. The trajectory visualization of each subnetwork helps us to understand whether they escape the current local basin. Even though discovered by the same ensemble method, their training trajectories are very different, proving the correctness of our proposal. For the comparison among different efficient ensemble methods, we have another two diversity measurements, e.g., functional disagreement and KL divergence, which are widely used in the ensemble literature [5,6].
>
> **Q9: "Section 5.4: Based on Fig. 4 there is a claim that the pattern of ensemble accuracy is highly correlated with the accuracy of the individual sub-network. I encourage the authors to support this with a quantifiable measure. Additionally, visually Fig. 4 does not seem to have the same pattern of evolution, wrt. sub-network accuracy or deviation in performance of sub-networks. Could the authors clarify the statement in the manuscript?"**
> - **Quantifiable measure.**  Encouraged by your comment, we added two widely used correlation measurements to support our claim: Pearson correlation and Spearman correlation. We use these two methods to measure the correlation between the mean accuracy of individual networks and the accuracy of their ensemble (higher values refer to higher correlation). The results are shared here. We can find that the individual accuracy and the ensemble accuracy are very highly correlated on CIFAR-100. Their correlation is relatively low On CIFAR-10 likely due to the almost saturated accuracy of the individual subnetwork (96%). Given such high accuracy, even the usually clear and strong correlation between the sparsity and test accuracy becomes relatively vague on CIFAR-10 in Figures 4a and 4b.
> |  Methods  |  WRN-28-10 CIFAR-10| WRN-28-10 CIFAR-10  |  WRN28-10 CIFAR-100| WRN28-10 CIFAR-100 |
> | ------------- | :-----------: |:-----------:|:-------------:|:-----------: |
> |                   |Pearson correlation |  Spearman correlation | Pearson correlation |  Spearman correlation|
> | EDST Ensemble |  0.459 | 0.257 | 0.979 | 0.829 |
> | DST Ensemble  |  0.332 | 0.319 | 0.972 | 0.812 |
>
> - **Pattern of evolution.**  According to our observations, the pattern of evolution is relatively similar in Fig. 4, i.e., as the sparsity decreases, both the accuracy of ensemble and subnetworks increases, especially for CIFAR-100. This behavior is very common in the literature of sparse neural networks. The minor exception might be that, different from CIFAR-100, the ensemble accuracy on CIFAR-10 does not fully monotonically improve as sparsity decreases (still, in general so). The reason is that the prediction accuracy of a single subnetwork on CIFAR-10 is already very high (96%). There is not much room for the ensemble to improve, and training noise might start showing impacts here.
>
> **Q10: "Section 5.5: The manuscript uses the term "Individual Tickets" which does not match the nomenclature used in Fig. 5."**
>
> - Thank you for pointing this out. We have adjusted them and will update the revision in a couple of days.
>
> **Q11: "Section 5.6: The adversarial robustness results for CIFAR-10 show that the dense model is more robust than the sparse models. Could the authors clarify their statements in Section 5.6?0"**
>
> - Thanks for your comments. We think this is a fair point and we adjusted our claim as something like “The results show that our proposed ensemble methods also bring benefits to OOD performance on CIFAR-10 and CIFAR-100 and adversarial robustness on CIFAR-100.”

---

> ### Author Response · Authors · 2021-11-18
> **Response to Reviewer 4rju (2/4)**
>
>
> **Q6:  "The difference in ECE values between the best and remaining methods across different models and datasets is unstable. Could the authors comment on the significance of these values and their changes?”**
>
> - Expected Calibration Error (ECE)[2], is a widely adopted metric to approximate the difference in expectation between confidence and accuracy of machine learning models (i.e., miscalibration). Specifically, it partitions predictions into M equally-spaced bins, and then calculates a weighted average of the accuracy/confidence discrepancy in each of these bins. Larger ECE values represent a worse match between confidence and accuracy.
> - Actually, our proposed methods perform quite well on the CIFAR datasets (the lowest ECE is achieved by them). However, DST-based Ensemble becomes less appealing on ImageNet, likely due to the extended training time (200 epochs) compared with the standard 90 epochs. The reason behind this setting is that, as we mentioned in the submission, the state-of-the-art DST methods can not match the accuracy of dense ResNet-50 on ImageNet with a standard training time (90 epochs). To guarantee sufficient accuracy for the individual subnetworks, we follow the state-of-the-art training setups used in [3] and train each sparse ResNet-50 for 200 epochs. However, the extended training time can result in stronger miscalibration as shown in [4]. It is worth to mention that on ImageNet also some of the baseline ensemble methods present some trend fluctuations (e.g., MIMO or BatchEnsemble) Nevertheless, our proposed methods achieve superior results in other preferable aspects, e.g., prediction accuracy, corrupted datasets, and training/inference efficiency.
>
> - In point of fact, the values of ECE are quite robust with different runs, as detailed below.
> |  Methods  |    |  ECE (WRN28-10 CIFAR-10)   |   |
> | ------------- | :-----------: |:-----------:|:-------------:|
> |                   | run 1 |  run 2  | |
> | DST Ensemble   | 0.010  |   0.010  |  |
> | EDST Ensemble |  0.014 |  0.012 |  |
> |                   | run 1 |  run 2  |  run 3 |
> Single DST network | 0.022 | 0.022 | 0.023
>
> **Q7: "A breakdown of the various labels/samples that cause the disagreement between various FreeTickets would be instructional. Applying KL divergence across intermediate feature embeddings could trace the source of variations and how they have an effect on various metrics."**
>
> - Thank you. We really like your suggestions and shared our findings below.
> - **Breakdown of disagreements.**  We choose the first 10 classes in CIFAR-100 and measure the disagreement between the subnetworks belonging to various methods on these classes respectively. The overall breakdown of disagreement is very similar to the diversity measurement results, i.e., DST Ensemble > Static Ensemble > EDST Ensemble > LTR Ensemble.
> |  Class |  apple  | aquarium  | fish | baby | bear | beaver | bed | bee | beetle | bicycle|
> | ------------- | :-----------: |:-----------:|:-------------:|:-----------: |:-----------:|:-------------:|:-----------: |:-----------:|:-------------:|:-------------:|
> | LTR Ensemble   | 0.027 | 0.063 | 0.140 |  0.170 | 0.157 | 0.070 | 0.127  | 0.110  | 0.027  | 0.030 |
> | EDST Ensemble   | 0.043   |  0.067  |  0.153  |   0.187  |  0.137   | 0.090  |  0.107   | 0.080  |  0.027  |   0.067 |
> | Static Ensemble   | 0.077   | 0.090   |0.220 |   0.180 |  0.227  | 0.157  | 0.120  | 0.097  |0.047  |  0.073|
> | DST Ensemble   |    0.057 |    0.080  | 0.227  |  0.277 |  0.230  |  0.137  | 0.160   | 0.133 |  0.053  |  0.093   |
>
> - **KL divergence of intermediate feature embeddings.** To trace the source of variations, we layer-wisely measure the KL divergence between the ensemble members of each ensemble method. Each line is the averaged KL divergence among M=3 subnetworks. The results are sampled in the table below, while in the revised paper are presented in a figure for completeness. The overall pattern between different ensemble methods matches the KL divergence we reported in the submission, i.e., DST>static>EDST>LTR. Moreover, it is interesting to observe that the KL divergence is quite high at early layers for the DST Ensemble and the Static Ensemble, which implies that the source of their high diversity might locate in the early layers.
> |  Layer Index|  1| 2| 3| 4| 5| 10| 15| 20| 25| 29|
> | ------------- | :-----------: |:-----------:|:-------------:|:-----------: |:-----------:|:-------------:|:-----------: |:-----------:|:-------------:|:-------------:|
> | LTR Ensemble   | 0.075  | 0.615 |  0.015   |0.115  | 0.053 |  0.070  | 0.024 |   0.002 |  0.011|   0.183|
> | EDST Ensemble   | 1.461 |  0.320   |0.136 |  0.125 |  0.082|    0.208  | 0.030  |  0.009  | 0.011  | 0.228  |
> | Static Ensemble   |  30.524 | 3.907 |  1.147  | 3.439   |1.123  |  1.623 |  0.142   | 0.0347 | 0.0296 |  0.421 |
> | DST Ensemble   |       60.045 | 15.848 | 0.667 |  23.599 | 1.411   |  2.461  | 4.210  |  3.065 |  3.362 |  0.405 |

---

> ### Author Response · Authors · 2021-11-18
> **Response to Reviewer 4rju (1/4)**
>
> **Q1: The term "cheap" pertaining to one of the key properties required of FreeTickets is left undefined until Pg. 4, Section 3.1. While the comparison to dense networks and their ensembles is fair from the perspective of performance, could the authors clarify if methods like static LTR ensemble and others are similarly "cheap" to DST and EDST? Ultimately, tying a quantifiable comparison of total FLOPs to calculate ensembles to emphasize the notion cheap would be preferable.”**
>
> - We fully agree that a quantifiable comparison of total FLOPs among various methods is important. We actually have provided the total training FLOPs (i.e., total FLOPs required to obtain all the ensemble members during training) of all methods in the original submission, which is notated as ‘# Training FLOPs’ in the penultimate column in Tables 1, 2, and 3. We use this overall training FLOPs to enable comparison among different ensemble methods. We can observe that the “cheap” ranking is: EDST Ensemble > DST Ensemble = Static Sparse Ensemble > LTR Ensemble.
>
> **Q2: "Could the authors clarify what $n^{(L)}$ refers to in Pg. 4, L. 2?"**
> - The $n^{(L)}$  in Pg. 4, L. 2 refers to the total number of neurons in the classifier/last layer. We checked for consistency and unified the notations.
>
> **Q3: "Could the authors discuss the different decision factors that influence the choice of $t_{ex}$? While M controls the amount of time each sub-network is fine-tuned, to what degree does $t_{ex}$ need to be explored so that a sufficient embedding space is learned?"**
> The value of $t_{ex}$ determines the amount of training time with a large learning rate (i.e., 0.1). The high learning rate is important for DST methods since it allows the newly grown weights (initialized as zero) to receive enough updates to survive in the next pruning iteration, guaranteeing the effectiveness of the parameter exploration. Otherwise, the newly grown weights will be pruned immediately and the exploration operation will be invalid. On the other hand, given the fixed training steps $t_{total}$ and ensemble member $M$, an overlarge $t_{ex}$ would cause $t_{re}$ to be too small, leading to unsatisfying performance. We empirically find $t_{ex}$ = 150 generally performs well on CIFAR-10/100.
>
> **Q4: "Pg. 6, Paragraph on Baselines, the last sentence mentions that the LTR ensemble has low diversity and prohibitive costs. A quantifiable measure or citation would help qualify this statement.**
>  - We indeed have provided the quantifiable measurement in the original submission in Tables 1,2,3, and 4. For convenience, we summarize the result here. The diversity and ensemble accuracy of LTR ensemble are much lower than our methods. Moreover, similar observations have also been shown in the prior work [1].
> |  Methods  | | | WRN-28-10 CIFAR-10| | | WRN28-10 CIFAR-100| |
> | ------------- | :-----------: |:-----------:|:-------------:|:-----------: |:-----------:|:-------------:|:-------------:|
> |                   |Training FLOPs|$d_{dis}$ | $d_{KL}$ | Acc|$d_{dis}$ | $d_{KL}$ | Acc|
> | LTR Ensemble   |1.75x |    0.026 |     0.056   |   96.2      |       0.111   |   0.185    |82.1|
> | EDST Ensemble |  0.61x| 0.031  |  0.073 |  96.3  |  0.126 | 0.237 |  82.2|
> | DST Ensemble  |  1.01x| 0.035  |  0.095 |  96.3   |  0.166 | 0.411  | 83.3 |
>
> **Q5:  "Could the authors clarify if the training FLOPs include the total number of forward passes and the FLOPs for each pass or they quantify the FLOPs required for a single pass only?"**
>
> - The reported training FLOPs in the table are the total number of FLOPs required to obtain all the subnetworks (free tickets) during training. It includes all the forward passes, backward passes, and the FLOPs for each pass that happened during training. Moreover, as the final obtained subnetworks of our methods are very sparse, the FLOPs required in the test phase are also much smaller than the dense ensemble. Please refer to our second answer to reviewer **hJ1v** for more results.

---

> ### Author Response · Authors · 2021-11-26
> **Response to Reviewer 4rju**
>
> Dear Reviewer 4rju,
>
> We thank you for your review time and constructive reviews!
>
> We would like to kindly remind you that the discussion period is ending in three days. We have provided detailed replies and new experiments to your comments. We hope to have a
> further discussion with you to see if our response solves the concerns.
>
> Given all the new experiments and our response, are you willing to reconsider your rating? Your support is very important to us and we greatly appreciate that!
>
> Best Regards,
>
> Authors

---

> ### Author Response · Authors · 2021-11-29
> **[Last two days reminder] Would you mind checking our response?**
>
> Dear Reviewer 4rju,
>
> Thanks for the comments! We provided a detailed reply to every concern in your comments. Here is a summary of our response.
>
> - We have highlighted the total training FLOPs used in our submission to quantify the “cheap” property.
> - We have explained the effect of t_ex.
> - We have provided a detailed explanation about the values of ECE and their changes.
> - We have provided a breakdown of the various labels/samples that cause the disagreement.
> - We have provided KL divergence of intermediate feature embeddings.
> - We have adjusted our statements in Section 5.6.
>
> We would sincerely appreciate it if you could check our response.
>
> Best Regards,
>
> Authors

---

### Official Review · Reviewer_wRp3 · 2021-11-02

**Correctness:** 3
**Technical Novelty And Significance:** 3
**Empirical Novelty And Significance:** 3
**Recommendation:** 5
**Confidence:** 4

**Details Of Ethics Concerns:**

None.

**Main Review:**

This paper pointed out an interesting and potentially useful way of training sparse DNNs bypassing the difficulty of optimizing sparse DNNs, i.e. by choosing to ensemble weakly optimized multiple sparse subnets, instead of post-training compression or lengthy DST to reach a single subnet.

Major concerns:

- Inference flops.  From the presentation of the results, I do not understand how emsembling $M$ sparse subnets would be efficient at inference time.  Of course at a certain high level of sparsity, the flop count is going to be lower than the dense baseline, but how can it beat a single subnet at the same sparsity, such as one trained with RigL, in flop count?  Even though the training flop count for a single subnet by DST is much higher, the $M\times$ lower inference flop count would still be desirable in deployment.  If I understand correctly, a fair comparison (in the sense of equal inference flop count) should be between an ensemble of $M$ sparse subnets at sparsity $s$, and a single sparse subnet trained by DST at sparsity $1-M(1-s)$. However, the baselines for comparison are other ensembles instead of single nets.  Given the under-trained individual subnets (e.g. in Fig. 5b), single subnet even at the same sparsity might outperform a sparse ensemble.
- Dense ensemble baselines.  How does the sparse ensemble compare to an ensemble of thin, dense individual subnets at the same flop count?
- The paper presented hypotheses on the generalization performance of the sparse ensemble trained as such, which are interesting scientific questions, but did not provide convincing results or discussions to validate or disprove the hypotheses.  For example, it is not clear if *diversity*, i.e. disagreement between the subnets, is responsible for the generalization.  Counter-evidence exist, e.g. it has been shown that ensembling nets that are intentionally trained to agree with each other through knowledge distillation improves instead of hurts generalization (arXiv:1805.04770).  From a practical perspective, DST or EDST are by nature handicapped in producing diverse subnets, because the enclosing dense net architecture is still the same--in order to obtain diversity, wouldn't one ensemble individual nets of completely different architecture rather than sparse subnets of the same enclosing dense net?  Further, the ablation study on parameter exploration only reported diversity metrics, but how do they differ in generalization performance?  In Fig. 3, does ensembling a red, a green and a blue checkpoint give higher accuracy than ensembling say three red checkpoints?

Minor concerns:

- Writing.  Language needs substantial polishing for publication standard.  Too colloquial at places, e.g. "a tough nut to crack", and redundant and ungrammatical at others, e.g. "Currently, the current only way ...", "the training costs are far fewer ...", "exploring heavily a large fraction ...", etc.
- Some figure/table fonts are too small to be legible.
- Sparsely gated mixture of experts.  How does the sparse subnet ensemble compare with sparsely gated MoEs at the same training and inference flop counts?


**Summary Of The Paper:**

The authors proposed a method to train an ensemble of sparse DNNs by means of DST such as RigL which, within the same training flop budget, is reported to generalize better than the enclosing dense net or other efficient ensembles; it is also reported to have higher inference-time efficiency.  Further analysis was performed to understand the effectiveness of the sparse ensemble.


**Summary Of The Review:**

Though the paper presented interesting ideas, it unfortunately falls short of a systematic study to make a practically useful methodology, or of a compelling hypothesis test that shed light on underlying scientific questions.

---

> ### Author Response · Authors · 2021-11-18
> **Response to Reviewer wRp3 (2/3)**
>
> **Q3: "The paper presented hypotheses on the generalization performance of the sparse ensemble trained as such, which are interesting scientific questions, but did not provide convincing results or discussions to validate or disprove the hypotheses. For example, it is not clear if diversity, i.e. disagreement between the subnets, is responsible for the generalization. Counter-evidence exists, e.g. it has been shown that ensembling nets that are intentionally trained to agree with each other through knowledge distillation improve instead of hurts generalization (arXiv:1805.04770)."**
>
> * **Hypothesis support**:  Many prior works on deep ensembles [1,2,3] have shown that the quality of an ensemble highly depends on the diversity of its members. And the diversity measurements used in our paper, i.e., prediction disagreement and KL divergence, have been widely used in the ensemble literature, such as BatchEnsemble [4], Hyper-batch Ensemble [5], and MIMO [6]. We clarify that we indeed have provided empirical results to support our hypothesis, throughout the paper. We summarize them here to clarify this question, for your convenience.
> * **Diversity**: We can observe a very strong correlation between the model diversity and the ensemble accuracy. PF Ensemble refers to pruning and fine-tuning here. We measured the Pearson correlation between model diversity and ensemble accuracy. The correlation coefficient (larger values refer to higher correlation) shows a very high correlation with values of 0.860 and 0.950 for CIFAR-10 and CIFAR-100, respectively.
> |  Methods  | |WRN-28-10 CIFAR-10| | | WRN28-10 CIFAR-100| |
> | ------------- | :-----------: |:-----------:|:-------------:|:-----------: |:-----------:|:-------------:|
> |                   |$d_{dis}$ | $d_{KL}$ | Acc|$d_{dis}$ | $d_{KL}$ | Acc|
> | Single DST |  0.026   |   0.056  |    96.2   |   0.111 |  0.185  |  82.1 |
> | EDST Ensemble |  0.031  |  0.073 |  96.3  |  0.126 | 0.237 |  82.2|
> | PF Ensemble     |  0.035  |  0.103 |  96.4   |  0.148 | 0.345 | 83.2 |
> | DST Ensemble  |  0.035  |  0.095 |  96.3   |  0.166 | 0.411  | 83.3 |
> * **Expressibility**: Typically, smaller sparsity leads to higher prediction accuracy along with higher expressibility for sparse subnetworks [7].
> As shown below, the accuracy of DST Ensemble consistently increases as the sparsity decreases. We measured the Pearson correlation between the single model's accuracy and ensemble accuracy.
> The correlation coefficient again shows a very high correlation with values of 0.972 for CIFAR-100.
> |WRN-28-10 CIFAR-100 | | | | | | |
> | ------------- | :-----------: |:-----------:|:-------------:|:-----------: |:-----------:|:-------------:|
> Sparsity |                                  0.95  |   0.90  |  0.80  |  0.70 |  0.60 |   0.50     |
> DST Ensemble accuracy |         82.60 | 83.15 | 83.26 | 83.42 | 83.25| 83.36 |
> Single model accuracy |          79.72 | 80.71 | 81.10 | 81.20 | 81.20 | 81.43 |
> * **Counter-evidence on BANs**. While we emphasize diversity is important, it's not equivalent to "combining vastly different models". In fact, there is inherent agreement among our sparse members (e.g., the prediction agreement (1-D_dis) of our methods can be around 97% for CIFAR10 and 85% for CIFAR100). We will clarify this prerequisite in the paper.
>
> - In the BANs paper (arXiv:1805.04770), the authors studied a very different context: KD-train several generations of dense students, in sharp contrast to our setting (sparse training in one training pass for EDST). In their setting, diversity is embedded due to each time initialized from a different random seed and other randomness during different passes of training.
>
> - So to summarize: both works balance between diversity and commodity among ensemble members, just in different forms. Their diversity is from multiple training passes' randomness. The diversity of EDST Ensemble is from the sparse connectivity change in one training pass and the diversity of DST Ensemble is from the randomness of different runs together with the different sparse connectivities. We are happy to add this discussion in the revision, and please note our unique benefit of sparse training efficiency ( multiple computationally light training runs for the DST Ensemble and one pass to get them all for the EDST Ensemble ).

---

> ### Author Response · Authors · 2021-11-18
> **Response to Reviewer wRp3 (1/3)**
>
> We thank the reviewer for the high-level and detailed feedback. In the following, we address the concerns raised by the reviewer. If some responses are unclear or you wish additional changes, please let us know.
>
>  &nbsp;
>  &nbsp;
>
> **Q1: "Inference flops. From the presentation of the results, I do not understand how ensembling M sparse subnets would be efficient at inference time. If I understand correctly, a fair comparison (in the sense of equal inference flop count) should be between an ensemble of **M** sparse subnets at sparsity **s**, and a single sparse subnet trained by DST at sparsity **1-M(1-s)**.**
>
> * We want to emphasize that the central goal of our paper (agreed by other reviewers) is to develop efficient ensemble methods that enable promising ensemble performance while reducing the large training/inference cost required by the traditional deep ensemble. The proposed ensemble methods enable to achieve even better performance than the strong dense ensemble while being even more efficient to train than a single dense model.
> Since our methods can even outperform the strong baseline, i.e., dense ensemble (much more performative than one single dense network), the superior performance is beyond the reach of any single network (either sparse or dense).
> Following your suggestion, we run a single Wide ResNet28-10 (WRN28-10) trained by DST at sparsity 1-3(1-0.8) = 0.4 on CIFAR and ResNet-50 (RN-50) at sparsity 1-2(1-0.8) = 0.6. The results are shared here. The accuracy [%] achieved by one single subnetwork is clearly lower than the accuracy achieved by our proposed methods.
> |  Methods  | WRN28-10 CIFAR10 | WRN28-10 CIFAR100| ResNet-50 ImageNet |
> | ------------- | :-----------: |:-----------:|:-------------:|
> | Single DST | 95.9  |  81.2 | 76.1 |
> | Dense Ensemble| 96.6 | 82.7 | 77.5 |
> | EDST Ensemble |  96.3  |  82.2 | 76.9 |
> | DST Ensemble| 96.3   |  83.3 | 78.3 |
>
> **Q2: "Dense ensemble baselines. How does the sparse ensemble compare to an ensemble of thin, dense individual subnets at the same flop count?"**
>
> * As we mentioned above, our methods can match or even outperform the strong baseline dense ensemble while using much fewer training FLOPs, so that the thin, dense ensemble would not be competitive here. We hope our answer to the first question has also clarified potential confusion.

---

> ### Author Response · Authors · 2021-11-26
> **Response to Reviewer wRp3**
>
> Dear Reviewer wRp3,
>
> Thanks a lot for your constructive review!
>
> We would like to kindly remind you that the discussion period is ending soon. We provided a detailed reply to every concern in your comments. Would you mind checking them and seeing if they successfully addressed your concerns?
>
> Best Regards,
>
> Authors

---

> ### Author Response · Authors · 2021-11-29
> **[Last two days reminder] Would you mind checking our response?**
>
> Dear Reviewer wRp3,
>
> Thanks for the comments! We have tried our best to address all the concerns and provided new experiments to your concerns. Here is a summary of our response.
>
> - We provided new experiments of a single sparse subnet trained by DST with sparsity $1-M(1-s)$(Q1).
> - We summarized the empirical support of the hypothesis proposed in our paper about FreeTickets (Q3).
> - We provided the ensemble accuracy of the ablation study on parameter exploration (Q5).
> - We provided new experiments on the ensemble of three red points (Q6).
> - We provided discussions about the related work of BANs (Q3) and sparse MoE (Q7).
>
> We humbly expect you can check it and confirm whether our response has addressed your concerns.
>
> Best Regards,
>
> Authors

---

### Author Response · Authors · 2021-11-18
**Response to all reviewers**

We highly thank all the reviewers and area chairs for their detailed and insightful comments. We are glad that they found our paper to be novel and interesting and supported with strong results. We have added all the experiments suggested by the reviewers and provide our responses below. We are currently working on preparing a revision of our paper based on this discussion and the reviewers’ comments. We will submit this revised paper in the next few days.

---

### Author Response · Authors · 2021-11-23
**Revision**

We have uploaded a revision of the manuscript that includes all the promised experiments, discussions, and changes. We provide a brief summary here.


- Adding baselines of pruning and finetuning ensemble (Table 1 & 2), FGE, and Snapshot (Appendix E).
- Adding disagreement breakdown (Appendix G) and KL divergence across intermediate layers (Figure 3-left).
- Adding analysis of q (Appendix I) and analysis of regrowth methods (Appendix J).
- Adding correlation measurement between the accuracy of individual subnetworks and the ensemble (Appendix H).


- We have tried our best to polish our paper.
- Adding more elaborate descriptions of EDST Ensemble (Section 3.3).
- Adding a longer description of baselines in the related work.
- We modify Figure 1 for compactness and legibility.
- We have adjusted the abstract and introduction to focus on the efficient ensemble.


We would like to thank all reviewers for their time and feedback again, and we really hope to have a further discussion to see if our response solves the concerns. We would be more than happy to provide more information or clarification, should it be necessary.

---

### Author Response · Authors · 2021-11-30
**Pre-decision: Summary of updates from Authors**

Dear AC and all Reviewers,


We sincerely appreciate AC and all reviewers’ time and insightful comments, which help a lot in further improving our paper. We are so glad that the merits of our work have been unanimously recognized by reviewers **4rju**, **HPxD**, and **hJ1v**, who gracefully responded to us and re-assessed our work more positively.

We regret that we are unable to engage reviewer **wRp3** into any discussion during the 3-week window, after our repeated inquiries. We however believe our detailed response should have completely resolved all her/his concerns, point by point. We hope the reviewer could take a close look at our responses as well as other active discussions.

To give everybody a refresher on the additional experiments and clarifications we conducted to address the raised concerns, here is a summary of our responses:

 - **[Inference flops; comparison with a single subnet at sparsity 1-M(1-s)]**. The goal of our paper is to develop efficient sparse ensemble methods (agreed by other reviewers) that enable promising ensemble performance while largely reducing the training/inference costs.  Our methods can even outperform the strong baseline, i.e., dense ensemble, with only 1/4 inference flops of the latter. In this case, any single network would not be competitive here. We have added a single DST subnet at sparsity 1-M(1-s), whose accuracy is clearly lower than the accuracy achieved by our methods.
- **[Support/validation of the hypothesis on sparse ensemble]**. We clarify that we indeed have provided empirical results to support our hypothesis throughout the paper (also summarized in our response). Moreover, we have added additional quantifiable correlation measurements to further support our claim.
- **[Counter-evidence on diversity between the subnetworks in arXiv:1805.04770]**. We have provided a detailed discussion between the paper arXiv:1805.04770 and our work. To summarize: both works balanced diversity and commodity among ensemble members, yet in different forms; they also served completely different goals.
- **[The generalization performance in the ablation study]**. We have provided the generalization performance of the ablation study, which supports our claim as well.
- **[Ensemble of three red checkpoints]**. The ensemble of three red checkpoints leads to very limited accuracy gains due to the extremely low diversity, validating the effectiveness of our methods.

We believe that these new results, together with the ones from the original manuscript, should be able to well address all reviewer **wRp3’s** concerns.

Given the significant potential impacts of our work as unanimously recognized by reviewers **4rju**, **HPxD**, and **hJ1v**, we are confident that our work makes a valuable contribution of broad interest to the ICLR community. To summarize:
-  We introduce the concept of FreeTickets, a sparse ensemble framework that enables to match or improve over the dense baseline in the prediction accuracy and robustness, while having even fewer parameters and training FLOPs compared to a single dense model.
- We demonstrate for the first time that DST methods may not match the performance of dense training when standalone, but
can surpass the generalization of dense solutions (including the dense ensemble) as an ensemble, while still being more efficient to train than the single dense network. (quoting reviewer **HPxD**)

We greatly appreciate the time and effort of reviewers and AC again!

Authors

---

### Decision · Program_Chairs · 2022-01-20

**Decision:**

Accept (Poster)

**Comment:**

#### Summary

The goal of this work is to reduce the costs of inference in ensembled models by ensembling sparse models. The paper also aims to reduce the costs of training these ensembles as well. The proposed techniques (DST and EDST) each these goals, respectively.

#### Discussion

As noted by the reviewers, the paper is interesting and timely. The authors provided significant clarifications in the response that satisfied the reviewers' concerns. There is still significant room to revise the remaining points and polish the text of the paper for the camera-ready (I highly recommend proofreading from an individual who is not an author on the paper; there are still typos in the revised edits)

#### Recommendation.

I recommend Accept, due to the strengths above and the reasonably scoped remaining work to do going into the camera-ready.